**DOI: 10.1038/ncomms15477**　　**OPEN**

# Self-folding origami at any energy scale

Matthew B. Pinson[1,*], Menachem Stern[1,*], Alexandra Carruthers Ferrero[1,2], Thomas A. Witten[1], Elizabeth Chen[3] & Arvind Murugan[1]

Programmable stiff sheets with a single low-energy folding motion have been sought in fields ranging from the ancient art of origami to modern meta-materials research. Despite such attention, only two extreme classes of crease patterns are usually studied; special Miura-Ori-based zero-energy patterns, in which crease folding requires no sheet bending, and random patterns with high-energy folding, in which the sheet bends as much as creases fold. We present a physical approach that allows systematic exploration of the entire space of crease patterns as a function of the folding energy. Consequently, we uncover statistical results in origami, finding the entropy of crease patterns of given folding energy. Notably, we identify three classes of Mountain-Valley choices that have widely varying 'typical' folding energies. Our work opens up a wealth of experimentally relevant self-folding origami designs not reliant on Miura-Ori, the Kawasaki condition or any special symmetry in space.

[1] Physics and the James Franck Institute, University of Chicago, Chicago, Illinois 60637, USA. [2] Physics department, University of Puerto Rico, Rio Pierdas Campus, San Juan, Puerto Rico 00931, USA. [3] SEAS (School of Engineering and Applied Sciences), Harvard University, Cambridge, Massachusetts 02138, USA. * These authors contributed equally to this work. Correspondence and requests for materials should be addressed to A.M. (email: amurugan@uchicago.edu).

Programmed instabilities and weak spots have emerged as a powerful tool to design a unique preferred deformation mode into mechanical structures[1,2]. Such mechanisms are attractive in actuators[3,4], meta-materials[5], art, architecture[6,7], robotics[8] and other applications at different length scales, because mechanisms require minimal control at the time of deployment; as seen in folding chairs or unfolding umbrellas, the designed deformation is a unique one-dimensional path in configuration space through which the structure is naturally guided under any external force. Mechanisms[9], even more so than marginal structures[10], are delicately poised at the boundary between being rigid and floppy. Despite much recent interest in large extended mechanisms[6,11–15] and some critical contributions towards the same[6,16–20], most work has focused on deformations with high symmetry, and the space of designed disordered deformations remains largely unexplored.

A prominent and ancient example of designed deformation is origami. In particular, rigid origami is the study of stiff sheets that do not bend except at the prescribed creases[6]. If creases are placed at just the correct angles relative to each other, the sheet as a whole has exactly one allowed deformation in which all the creases fold at the same time. Such sheets can be described[21] as self-folding because the allowed mode will be actuated by almost any applied force; there is no need to precisely tailor the folding forces. While a general origami pattern might have several folding motions, a self-folding pattern will have a unique extended motion that requires less energy than all others.

However, even in this well-studied area, most known examples of self-folding crease patterns are in fact rigidly foldable (that is, foldable at precisely zero energy cost). With the exception of some influential works discussed below[6,16–20], rigidly foldable crease patterns are often periodic structures made of repeating units, such as Miura-Ori and its derivatives[22,23]. Further, origami design has often been limited to the Mountain-Valley (MV) pattern implicit in Miura-Ori[6,11,24]. Many such studies of rigid origami have also been restricted to so-called flat-foldable or near-flat-foldable vertices[6,25] (that is, patterns in which all creases fold to angle $\pi$ simultaneously); the flat-foldability restriction on angles in a crease pattern leads to dramatic algebraic simplifications in rigidity calculations. As a result, Miura-Ori derivatives are often rigidly foldable, with the stiff sheet between creases (that is, the 'faces') not bending at all when the creases are folded.

Restricting study to the rigid foldable patterns with no face bending misses a larger space of near-perfect mechanisms, in which face bending or energy cost of actuation can be made arbitrarily small. Understanding the full space of crease patterns as a function of folding energy is also crucial for self-folding origami applications[13], since applications vary widely in material stiffness and actuation torques (or energies) available. For example, folding a structure made of stiff plates connected by shape-memory polymer hinges[26,27] that provide low-actuation torques might require nearly rigid foldable patterns; but using shape-memory alloys[28] or ionic electroactive polymer[13] hinges that provide higher torques would allow use of less foldable patterns as well. Similarly, one might wish to prevent accidental deployment of a self-folding hydrogel capsule[29] due to small pH fluctuations, necessitating less foldable patterns.

Surprisingly little is known about general self-folding origami patterns that are not exactly rigidly foldable. Important contributions include Huffman's work[17] on general $n$-valent vertices and Tachi's simulation scheme of origami patterns[6]. Wu and You [20] introduced analytic methods to analyse motions of multi-vertex patterns, extending Belcastro and Hull's condition for testing rigid-foldability[18] for non-flat foldable patterns. Tachi went beyond rigid foldability for general patterns by establishing design principles for first order foldability[16,19].

Energy scale-dependent origami design and statistical properties of typical patterns are the basic building blocks needed for a physically motivated theory of origami[15], relevant to both natural[4] and synthetic[13,30] systems. In this work, we present a systematic exploration of the space of self-folding crease patterns as a function of folding energy by solving equations in sequence. We further show that MV choices strongly affect foldability, for example, 62% of all MV choices account for 10% of highly foldable patterns. Finally, we find an entropy–energy relationship quantifying the number of crease patterns with given folding energy, describing how many more crease patterns become available for a given increase in available actuation energy, for example, in active hinges[13].

## Results

**Vertex transfer function.** As in past work[6,11,23], we study patterns made of general 4-vertices, like those shown in Fig. 1, since vertices with three or fewer edges are completely rigid while vertices with more than four edges are too soft (that is, have multiple continuous degrees of freedom). Assuming the angles $\theta_{12},\theta_{23},\theta_{34},\theta_{41}$ between creases of the vertex add to $2\pi$, we note two primary facts about generalized 4-vertices studied earlier[23]; three out of the four creases must fold in a common orientation (say, Mountain, black in Fig. 2a) with the final odd-one-out crease folding the other way (Valley state, red). The final odd-one-out crease can be any one of the two creases whose neighbouring angles add to less than $\pi$ (ref. 23) (Supplementary Fig. 1). Once the discrete odd-one-out choice in MV has been made, a 4-vertex has exactly one folding degree of freedom (Fig. 2a); the folding angle $\rho_i$ at any crease $i$ completely determines any other folding angle $\rho_j$. For two chosen adjacent creases, we symbolically write,

$$\rho_1 = T(\rho_2; \{\theta\}) \qquad (1)$$

where $\{\theta\}$ are the four in-plane angles between creases.

For small fold angles $\rho_i$, we can linearize the above relationship and write

$$\rho_1 \approx R(\{\theta\})\rho_2 + O(\rho_2^2). \qquad (2)$$

$R$'s determine the mechanical advantage and dynamic range of folding angles at a vertex.

Similar transfer functions have appeared in the literature over the years[6,9,15,17,31–33]. We emphasize that the transfer functions $T$, $R$ depend on the MV configuration at the vertex[23]. Explicit forms of $T$, $R$ for general 4-vertices, including their MV dependence, are presented in Supplementary Note 1.

**Loop equation and tuneable stiffness.** While a single 4-vertex (Fig. 2a) always has one degree of freedom, 4-vertices linked to form a quad are generically rigid. In fact, the number of folding degrees of freedom (that is, 12 folding angles $\rho_i$) exactly matches the number of constraints relating these folding angles (three at each vertex). Hence a generic quad has, at best, a discrete set of folded states—the folding motion between such states will generically involve face bending or other such violation of constraints.

Thus, smooth folding motions (modes) require fine tuning of the in-plane angles at each vertex (design parameters). An intuitive way to understand the fine tuning required is to write a consistency loop condition for a fold angle $\rho$, say that of $AD$ (see Fig. 2b), transported around the quad (that is, forming a closed loop),

$$\rho = T^D\left(T^C\left(T^B\left(T^A(\rho)\right)\right)\right) \qquad (3)$$

This nonlinear loop equation needs to be satisfied as a function of $\rho$, not just at particular values of $\rho$, in order to have a smooth folding mode. Taylor expanding the right hand side and subtracting $\rho$,

$$0 \equiv f_1\rho + f_2\rho^2 + f_3\rho^3 + \dots \qquad (4)$$

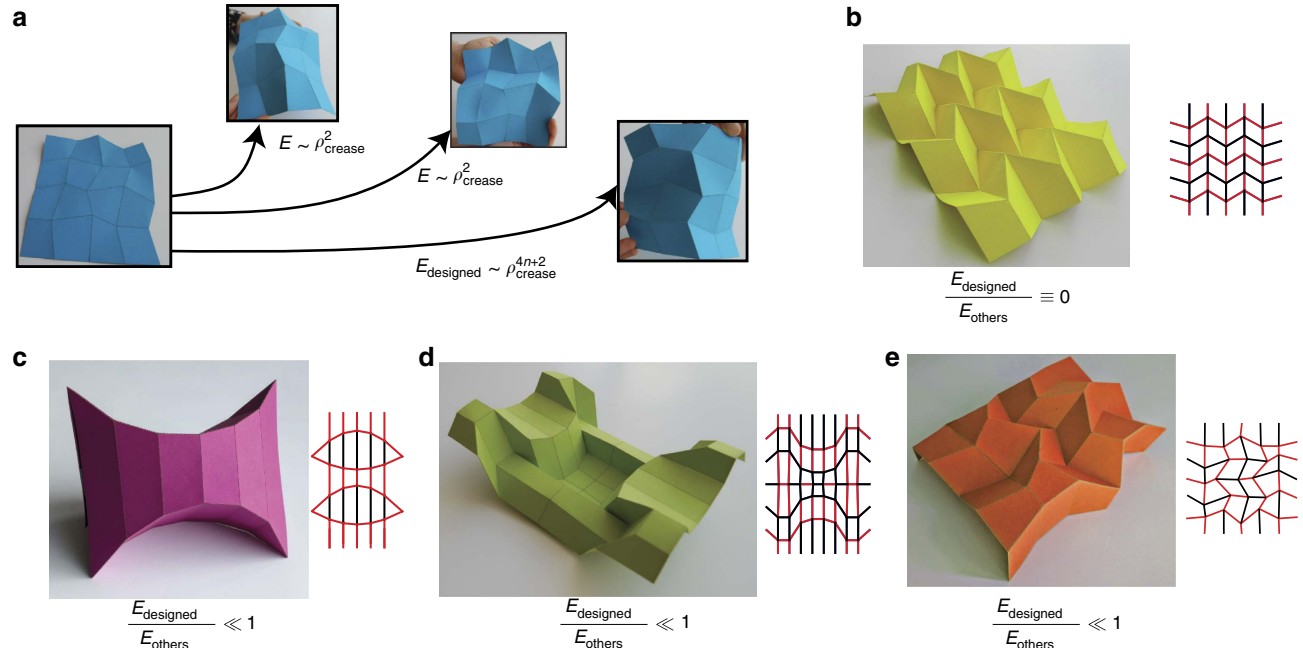

**Figure 1 | Designing self-folding origami.** (**a**) Forces applied to a 'self-folding' sheet will preferentially actuate the one pathway designed to have significantly less face bending than the other two pathways shown (that is, designed to have $E_{designed}/E_{others} \ll 1$). (**b**) The celebrated Miura-Ori pattern is a special highly symmetric pattern with $E_{designed}/E_{others} \equiv 0$. (**c–e**) In this work, we study a larger space of experimentally relevant crease patterns by going beyond rigidly foldable symmetric patterns. The folding energy scale of such patterns can be made as small as needed in a systematic manner; $E_{designed} \sim \rho_{crease}^{4n+2}$ where $\rho_{crease}$ is the median crease folding angle, and $n$ the number of solved loop equations that are derived here. Patterns in **c–e** are geometrically distinct from traditionally studied limits (Kawasaki vertices, Miura-Ori Mountain-Valley choice). These patterns solve exactly only one (**d,e**) or two (**c**) loop equations.

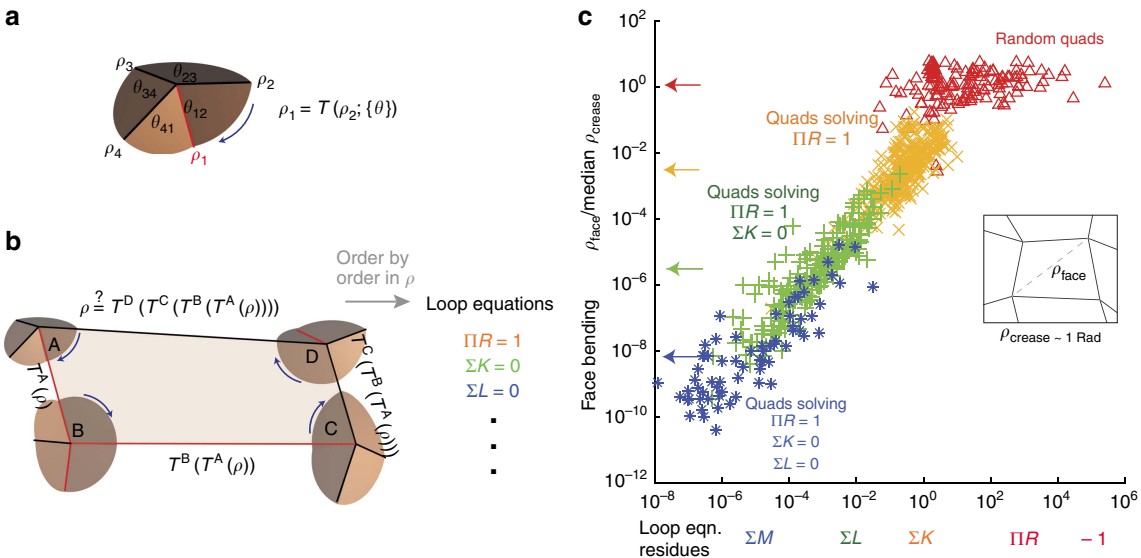

**Figure 2 | Loop equations uncover folding modes of variable face bending over orders of magnitude.** (**a**) Folding angles $\rho_1$, $\rho_2$ of adjacent creases at a 4-vertex are related by a transfer function $T$, determined by in-plane angles $\theta$. (**b**) For a quad to be foldable, a fold angle $\rho$ of an edge must return as $\rho$ when transported around the loop using transfer functions $T^A$, $T^B$, $T^C$, $T^D$. For a smooth folding motion, the equation must be satisfied order by order in $\rho$. (**c**) Face bending can be dramatically reduced in a controlled manner by solving loop equations in sequence. Random quads (red triangles), not designed to solve any loop equation, show face bending comparable to crease folding. Quads solving the first loop equation $\Pi R = 1$ (orange × s) typically have face bending $< 10^{-2}$ Rad. We find that the residue of the highest loop equation not solved determines the extent of face bending; hence orange points and green points show the drop of face bending with decreasing $\sum K$ and $\sum L$ respectively.

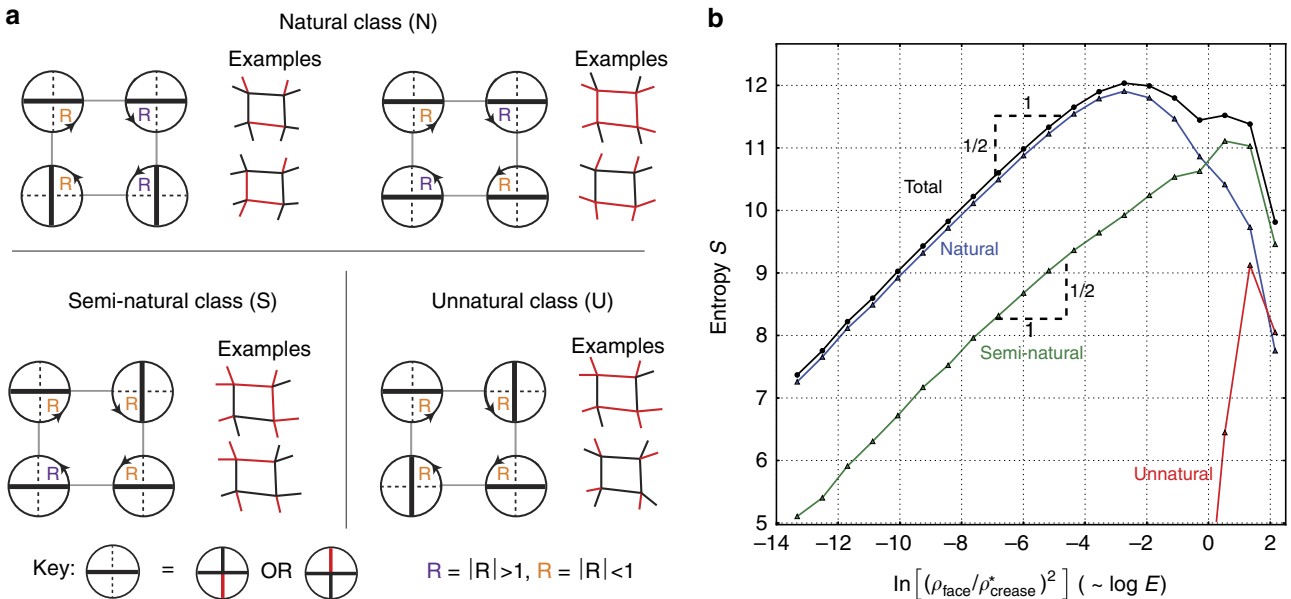

**Figure 3 | Mountain-Valley choices fall into three classes based on foldability of typical modes.** (**a**) For example, semi-natural MV configurations dictate that $|R| > 1$ at three vertices and $|R| < 1$ at one, which is statistically less compatible with $\Pi R = 1$ than natural configurations (two $|R| > 1$, two $|R| < 1$). (**b**) Consequently, when random modes of N, S and U types are sampled, U (unnatural) type modes tend to be much stiffer than S or N type. We sampled $10^6$ random modes, simulated folding and recorded their stiffness; we show the histogram binned by (log) face bending energy $\log \rho_{face}^2 / \rho_{crease}^{*2} \sim \log E$, which we call the entropy $S(E)$. ($\rho_{crease}^* \equiv$ median $\rho_{crease}$). Among soft patterns ($E < 10^{-1}$), 90% of random modes are of Natural MV type which account for only 6/16th of all MV configurations. S- and U-type modes dominate at high energies $E > 10^{-1}$. The histogram captures a statistical relationship between MV choices and foldability for 'typical' quad patterns and MV classes.

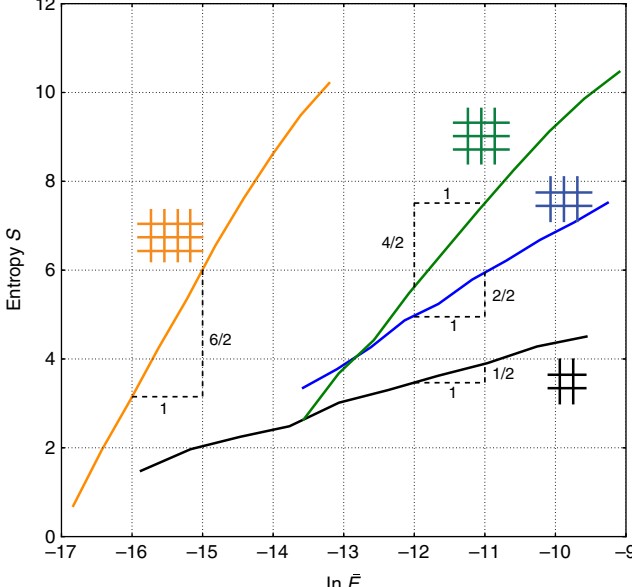

**Figure 4 | Entropy of crease patterns as a function of face bending energy.** We sampled $\sim 10^5$ random quad meshes made of $A$ quads (for $A = 1, 2, 4, 6$), folded them with the same fixed Mountain-Valley choice and noted the face bending energy per quad $\bar{E} \equiv (1/A) \sum \rho_{face}^2 / \text{median} \rho_{crease}^2$. The entropy of patterns is $S = \frac{A}{2} \log \bar{E}$. $e^{S(\bar{E})} d \log \bar{E}$ is the number of random patterns in an energy interval $d \log \bar{E}$.). Thus the probability of finding a soft crease pattern $e^{S(\bar{E})} d \log \bar{E} \sim \bar{E}^{\frac{A}{2}}$ in a random ensemble diminishes exponentially with mesh size $A$ (for fixed $\bar{E}$) but only as a power law in energy $\bar{E}$ (for fixed $A$).

where $f_1 = \Pi R - 1$. Setting $f_i = 0 \ \forall i$ gives—potentially—an infinite set of equations for the design parameters (that is, in-plane angles $\theta$s)—to have a folding motion to all orders in $\rho$. Similar loop equations for lowest order foldability were derived by Tachi[16,19] earlier.

We can write the series of loop equations, defined term by term using the expansion of the transfer function of equation (4). The loop equations are computed explicitly in Supplementary Note 1, while here we write them symbolically as

$$
\begin{aligned}
\Pi R : & \ R^A R^B R^C R^D = 1 \\
\Sigma K : & \ K^A + K^B + K^C + K^D = 0 \\
\Sigma L : & \ L^A + L^B + L^C + L^D = 0. \\
& \vdots
\end{aligned}
\tag{5}
$$

As shown in Supplementary Note 1, $R^V$ is a property of in-plane angles at a single vertex V. $K^V, L^V, \ldots$ are products of functions of a single vertex V and of $R_{V'}$ at other vertices $V \neq V'$.

MATLAB Code to compute loop equations to arbitrary order is given as Supplementary Material. For a quad, we verified that the first five equations are independent. Combined with Tachi's earlier work[6] that discovered a six-parameter family of rigidly foldable quads with a special symmetry (flat foldability), our work suggests that only the first five-loop equations are fully independent (Supplementary Note 2), as each loop equation constrains one parameter of the 11d $\{\theta\}$ design space. Here we focus on exploring the full space of creases patterns as a function of foldability and MV choices.

When a quad does not satisfy all loop equations exactly, there is no perfect zero-energy mode. Allowing a single diagonal fold (Fig. 2c, inset) adds an additional degree of freedom and thus

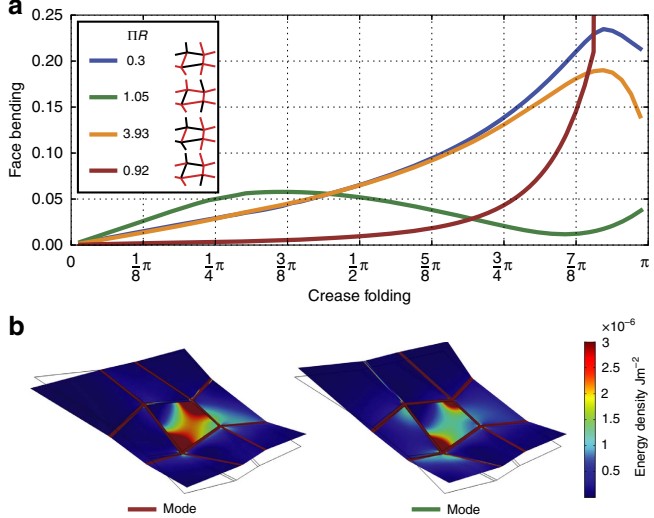

**Figure 5 | Face bending for large folding angles and in finite element simulations.** (**a**) Face bending for a quad when folded along different Mountain-Valley modes. While stiffness for small folding angles less than, say, $\pi/2$ is predicted by the loop equation residue $|\Pi R - 1|$, the initially soft red mode becomes stiffer than others at large folding angles. The other three modes show non-monotonic bistable face bending. (**b**) Finite element simulations (COMSOL Multiphysics) of a 2D plate model of the same quad when folded along the red and green modes with a fixed magnitude of folding force. The associated face deformation elastic energy is localized to a diagonal furrow as predicted for thin plates[34]. (Material model: melamine resin, elastic modulus 6 GPa, density 1,800 kg m$^{-3}$, thickness of plates is $\sim 1/1,000$ of width.)

allows any augmented quad (a quad with an additional face diagonal crease) to fold. Measuring the angle $\rho_{face}$ of a freely folding diagonal is a proxy for the face bending energy in the presence of a stiff face (Supplementary Fig. 2).

We note that stretching energy in thin sheets scales the same way with $\rho_{face}$ as bending energy due to a virial theorem[34,35], but is expected to be considerably smaller[34,35]. Further, for thin sheets, bending strain is much larger than stretching strain in low-energy configurations[34,35]. Hence, in the following, we model both the energy and geometry by considering only face bending. We later check the validity of this thin sheet approximation using finite element simulations in COMSOL (Fig. 5 and Supplementary Fig. 2). Thickness in real application varies, for example, 0.05 mm thick NiTi sheets of width 50 mm for stents[28] to 1-μm-thick GaAs sheets of width 100 μm (ref. 36) for optically actuated mirrors.

To study the relationship between loop equations and face bending quantitatively, we generated random quadrilateral patterns with random MV assignments and used them to solve loop equations order by order using gradient descent. We then added a crease along the face diagonal (Fig. 2c inset), simulated folding of each augmented quad from the unfolded state through small folding angles. In this way, we find,

$$\rho_{face} = a_1 |\Pi R - 1| \rho_{crease} + a_3 |\Sigma K| \rho_{crease}^3 + a_5 |\Sigma L| \rho_{crease}^5 + \dots \tag{6}$$

where $\rho_{crease} > 0$ is the median crease folding angle and the coefficient $a_i$ depend on the details (that is, $\theta_{ij}$) of the quad. Thus, as noted in Fig. 1a, the energy required to actuate the designed mode,

$$E_{designed} \sim \rho_{face}^2 \sim \rho_{crease}^{4n+2},$$

drops rapidly with the number $n$ of the exactly satisfied loop equations in the hierarchy. Applying the loop equation hierarchy to different seeds of pattern designs allows discovery of soft foldable patterns devoid of symmetries or order in space (Fig. 1c–e). Such soft patterns may have interesting mechanical properties that distinguish them significantly from the well-studied Miura-Ori pattern (Fig. 1b).

Remarkably, the relationship between face bending $\rho_{face}$ and $n$ strongly persists even if face bending is determined after folding to a large angle $\rho_{crease} \sim 1$ Rad. As shown in Fig. 2c, the loop equations when solved in sequence provide a controlled and systematic reduction in face bending over nine orders of magnitude. Solving each successive equation reduces face bending by a factor of $\sim 10^2$. In addition, the residue of the leading loop equation 'not' exactly solved is highly predictive of face bending. Thus the value of $\Sigma K$ is predictive of face bending for quads that solve $\Pi R = 1$ (orange $\times$), while $\Sigma L$ is predictive of face bending for quads that solve $\Pi R = 1$ and $\Sigma K = 0$ (green $+$) and so on.

Equation (5) thus provides a simple design principle for exploring the crease patterns at any chosen folding energy scale over many orders of magnitude; one simply solves the hierarchy of loop equations to the extent needed. Note that if the creases themselves have non-zero folding energy (for example, due to finite thickness), the folding energy $E_{designed}$ would be bounded from below by such an energy scale; crease patterns cannot be made softer than the intrinsic stiffness of individual creases.

**Mountain-Valley choice strongly affects foldability.** The loop equations explicitly depend on the MV choices around the quad. The equations can be defined for any given MV choice, opening up the full space of origami patterns. Almost all work-to-date on origami is based on Miura-Ori's MV choice. In the following, we show that different MV choices lead to different typical foldability in a statistical sense.

We find that some MV choices are intrinsically more conducive to solving the loop equations than others. Hence we can categorize MV choices by foldability classes. To define these classes precisely, note that at each vertex, one we can define the 'broken' direction to be the two longitudinal creases whose MV states differ (key in Fig. 3a). The two creases in the orthogonal 'unbroken' direction have the same MV state. The crucial observation is that the creases in the unbroken direction 'typically' fold more than the broken creases; hence $|R_{ij}| \approx |\rho_i/\rho_j| < 1$ if $i$ is in the broken direction and $j$ unbroken (see equation (2)).

Intuitively, some MV choices tend to make $|R| > 1$ at two of the 4-vertices around a quad and $|R| < 1$ at the other two. These natural MV assignments (Fig. 3a) are most easily compatible with $\Pi R = 1$ (equation (5)). Semi-natural MV patterns have $|R| > 1$ at three vertices and $|R| < 1$ at the fourth (or vice-versa). Finally, unnatural quads have all four $|R| > 1$ (or $|R| < 1$); the in-plane $\theta$ angles of such a quad must be fine-tuned to be foldable.

The class of a MV choice thus determines how easy it is to solve the first loop equation $\Pi R = 1$. Random quads with Unnatural MV choice are far less foldable than natural or semi-natural types (Supplementary Fig. 3).

To quantify this statement, we sampled $\sim 10^6$ random quads by displacing the vertices of a regular square lattice randomly and independently. We then simulated folding each of these quads with a random folding torque to obtain a folding mode with max $\rho_{crease} \sim 1$ Rad; we noted the resulting MV data as well as the resultant face bending $\rho_{face}$ for each mode. The histogram, binned by face bending energy $\log E = \log \rho_{face}^2 + \text{const}$ is shown in Fig. 3b; we define the entries of this histogram to be the entropy $S(E)$ since $\int_{E_1}^{E_2} e^{S(E)} d \log E$ gives the number of modes in our

random ensemble in the energy range $E_1 - E_2$ ($S(E)$ is defined only up to an additive normalization constant).

First, in Fig. 3b, we note that the total entropy follows a simple law up to quite stiff modes, $S(E) = \frac{1}{2} \log E$ that we explore further below. We also see that 90% of modes softer than $E \sim 10^{-1}$ are accounted for by Natural MV configurations, even though such configurations only account for 6/16 of all MV choices. Most of the remaining 10% of soft modes are accounted for by semi-natural configurations (8/16 of all choices). Among stiff modes $E > 10^{-1}$, the situation is reversed and semi-natural and unnatural configurations form a majority.

Thus, in addition to opening the door to arbitrary MV choices, our work suggests previously unnoticed MV classes that qualitatively differ in their typical foldability. Such widely varying entropy of MV classes suggests important lessons in design; natural configurations can be expected to be more forgiving of error in laying out creases while unnatural configurations need to be highly fine-tuned to be foldable.

**Entropy–energy relationship for large crease patterns**. We have seen that a quad's folding mode can be made arbitrarily soft by solving a series of loop equations; but soft modes are rarer than stiff modes. In the following section we generalize these considerations to large origami meshes.

We sampled crease patterns made of $A$ quads by displacing the vertices of a regular lattice with $A$ inner faces plaquettes randomly and independently, in much the same way as for the quad above. We folded the resulting crease pattern using a torque that selects chosen MV data until $\max(\rho_{crease}) = 1$ Rad and recorded the resulting face bending $\rho_{face}$ on each quad. We then made a histogram of $(1/A) \sum \rho_{face}^2 / \text{median } \rho_{crease}^2 \sim E/A \equiv \bar{E}$ from a large sampling of such lattices of different sizes; see Fig. 4.

We again define the entropy of crease patterns (now for a given MV) as the logarithm of the above histogram. We find that this entropy is extensive in pattern size $A$ and has a simple form,

$$S(\bar{E}) = \frac{A}{2}\left(\log\frac{\bar{E}}{E_0}\right) + \dots \qquad (7)$$

where $\bar{E}$ is the (intensive) face bending energy per quad, the ellipsis represent sub-leading corrections in $A$ and $E_0$ is a constant discussed later. By construction, $e^{S(\bar{E})} d \log \bar{E}$ is the number of crease patterns of chosen MV with folding energy within an interval $d \log \bar{E}$ around $\bar{E}$.

We can understand the extensive scaling of entropy $S$ with $A$ and the $\log E$ dependence using the loop equations. As seen in Fig. 2c, for a single quad, face bending energy is simply related to loop equation residues, for example, $\bar{E} \sim \rho_{face}^2 \sim (\Sigma K)^2$ for the green points. On the other hand, we find that the fraction of quads in our random ensemble with loop residue less than $|\Sigma K|$ is simply proportional to $|\Sigma K|$; this is because patterns in our random ensemble appear to be uniformly distributed in their residues. Hence the total number of patterns of energy less than $E$ scales as $\sqrt{E}$. Setting $\int_0^E e^{S(E)} d \log E \sim \sqrt{E}$ (by our definition of entropy), we find $S(E) = \frac{1}{2} \log E$. (Note that the energy is not quite linear in the residue of the first loop equation $\Pi R - 1$ in Fig. 2c which is reflected in Fig. 3b as well.)

For large quad meshes, the above arguments apply to each quad since we need to solve loop equations independently for each quad in order to make softer patterns. For example, imposing a loop equation now removes $A$ times as many design variables. Hence we find $S(\bar{E}) = (A/2) \log \bar{E}$.

Finally, note the $\log E$ dependence for large lattices in equation (7) is expected to break down near an energy scale $E_0$ (the $10^5$ samples generated here were not sufficient to probe this

scale). In particular, Tachi's results[6] on rigid foldable patterns imply an entropy of $\sqrt{A}$ at zero energy.

Our investigations of entropy of folding modes as a function of energy connects to earlier work on crumpling transitions[37]; consider a sheet with a thermally variable crease pattern held at fixed temperature. At high temperatures, if the entropy of stiff modes is sufficiently high, the sheet might crumple for entropic reasons, even if energetically disfavoured. Earlier analytic approaches were restricted to the entropy of rigid-foldable modes on regular lattices[38,39] while our work is off-lattice and has a continuum energy $E$. Our entropy $S(E)$, based on quad meshes, only grows logarithmically in energy and hence does not show a first order transition.

The entropy–energy relationship in equation (7) has theoretical and practical implications. In particular, the probability of a random pattern (when folded with a fixed MV) being softer than energy $E$ decreases exponentially with mesh size $A$ but only as a power law with $E$.

Such results are useful in understanding the trade-off between energy scales and design freedom. Self-folding origami applications vary greatly in the energy $E_{material}$ needed to bend an uncreased face to a given angle, for example, compare a Young's modulus of $\sim 10^3$ Pa for hydrogels[29] to $\sim 10^7$ Pa for NiTi alloy in origami stents[28]. Similarly, actuation mechanisms for active hinges are diverse, including electric[13], optical[36], thermal[28] and chemical (pH) (ref. 29) methods. Hence, the actuation energy $E_{actuation}$ provided by active hinges (defined as work done by hinges during folding to 1 Rad) can vary widely, for example, compare torques of $\sim 6 \cdot 10^{-3}$ Nm in 30 mm-long shape-memory polymer hinges[26,27] to $5 \times$ or $400 \times$ that torque in ionic electroactive polymers or shape-memory alloys respectively[13].

Taken together, $E_{actuation}/E_{material}$ can vary greatly across applications. Our energy–entropy relation shows that the fraction of all patterns suitable for such an application is $\sim (E_{actuation}/E_{material})^{A/2}$ (for large $A$).

In addition, micron-scale applications might often have a design requirement to prevent inadvertent actuation due to uncontrolled noisy processes of a lower energy scale $E_{noise}$, for example, spontaneous temperature[28] or pH fluctuations in hydrogels[29] or random mechanical kicks. To avoid inadvertent actuation, the folding energy of patterns must be in the 'Goldilocks' zone between $E_{noise}$ and $E_{actuation}$. The fraction of all patterns in the 'Goldilocks' zone can be computed to be $(E_{actuation}/E_{material})^{A/2} - (E_{noise}/E_{material})^{A/2}$ for large $A$.

Equation (7) thus provides a basic guideline for how many more patterns become available if the actuation energy $E_{actuation}$ is raised, say, at the cost of higher power input[26] or if the energy of uncontrolled processes $E_{noise}$ is lowered.

While our results were derived for the simplest random ensemble, they can be adapted to other ensembles of patterns relevant to specific applications.

**Face bending along folding modes**. In this work, face bending was measured by augmenting the quad with a diagonal crease (Supplementary Methods) and then setting crease folding to one representative angle ($\sim 1$ Rad) in equation (6). It is reasonable to expect face bending behaves as nonlinear function of crease folding for large folding angles. In this section we study the face bending of folding modes for variable crease folding amplitudes.

In Fig. 5a, we show face bending as a function of crease folding for four different folding modes. We see that the loop equation residue $|\Pi R - 1|$ is a good predictor of face bending up to crease folding of $\sim \pi/2$ Rad. For larger folding, strongly nonlinear effects kick in; the initially soft red mode become stiffer than others

rapidly, while other modes show non-monotonic behaviour. Non-monotonic bistable behaviour has been seen before in experiments on highly symmetric flat-foldable patterns[11,22].

To visualize what face bending stresses might look like in a real material without a diagonal crease, we show results of a finite element simulation in COMSOL in Fig. 5b of two select modes from Fig. 5a. Unlike our simplified model, this simulation accounts for stretching, bending of all nine faces, finite thickness of the material and finite width of creases. While our simple diagonal crease model cannot capture the precise folding energies seen in the COMSOL simulation, we see that the bending stress is localized to a furrow along the diagonal, a result expected for thin sheets[34] (Supplementary Fig. 2 for more analysis and simulations). Further, when folded in COMSOL with a small but fixed folding force (note: not to fixed angle), the red mode shows higher stresses, implying that it is softer than the green mode, in agreement with our face bending model in Fig. 5a.

## Discussion

In this work, we have studied self-folding origami meshes as a function of folding energy, free from assumptions about MV data or symmetries such as flat-foldability. We found a design principle for self-folding patterns of arbitrary stiffness in terms of a series of loop equations applied to each quad in the pattern. These general patterns can exhibit diverse curvatures in three dimensions as compared to Miura-Ori. Related recent work[11] has achieved remarkable three dimensional structures by gradual modulation of Miura-Ori patterns on length scales much larger than the repeating unit cell; however, our work allows design on arbitrary length scales without being limited to any underlying repetitive motif.

MV data were found to greatly affect foldability. Natural MV types are typically much softer than Semi-natural or Unnatural types. This notion informs design decisions for soft self-folding modes, both as soft Natural modes are more numerous, but also as they are projected to be less prone to large stiffness fluctuations due to manufacturing errors.

We complemented these design principles with a statistical understanding of the space of all large quad meshes; we determined the total entropy of crease patterns of any given folding energy. Such a relationship tells us the number of patterns with a 'Goldilocks' folding energy that is lower than available actuation energy but high enough to prevent inadvertent actuation due to noisy uncontrolled processes.

While our work focused on quad meshes for concreteness, the loop equation hierarchy applies to any pattern made of arbitrary combinations of polygons, provided all vertices have valence four. We leave investigations of mechanisms with other topologies to future work.

In conclusion, understanding the space of crease patterns as a function of an energy scale combined with statistical results on the foldability of 'typical' patterns are crucial ingredients in developing a physically relevant theory of self-folding origami.

**Data availability**. Data supporting the findings of this study are available from the corresponding author on request.

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

## Acknowledgements

We thank Michael Brenner, Levi Dudte, Heinrich Jaeger, Anand Murugan and Sidney Nagel for discussions and feedback. E.C. acknowledges NSF MSPRF grant DMS-1204686. M.B.P. was supported by a ConocoPhillips fellowship from the American Australian

Association. We acknowledge NSF-MRSEC 1420709 for funding and the University of Chicago Research Computing Center for computing resources.

## Author contributions

All authors helped develop the theoretical tools. A.C.F. performed the experiment. M.S., M.B.P., A.M. carried out simulations, analysis and wrote the manuscript.

## Additional information

**Competing interests:** The authors declare no competing financial interests.

**Publisher's note**: 

