## [Peer Review File · Nature Communications]

Reviewers' comments:

Reviewer #2 (Remarks to the Author):

The paper under review describes a new approach to modeling rigidly-foldable structures whose lowest-energy motion is parameterized by a single degree of freedom. The approach used consists of characterizing loop equations for the folding angles around a polygon (quadrilaterals, in particular) of the crease pattern in a folded state. The first, second, third, etc. orders of these equations provide conditions for rigid foldability and may be used to describe the space of crease patterns that exhibit rigid-folding behavior. Furthermore, this approach is not reliant on the vertices of the crease pattern being flat-foldable, allowing more generality than is often present in such studies.

The paper is an interesting addition to the growing literature on rigid folds. The work appears to be correct and its methodologies sound. But the question is whether this paper's approach is novel and significant enough for publication in Nature Communications. I disagree with the authors on the novel aspects of their approach. For this reason, and this reason alone, I am not recommending the paper for publication in Nature Communications. But the editor(s) may choose to view the other novel aspects of this paper as enough to warrant publication. Therefore I will provide more detailed commentary below.

Specifically, the authors state in the third paragraph:

"... origami design thus far has mostly been limited to the mountain-valley (MV) pattern implicit in Miura-Ori. Further, most studies of rigid origami have been restricted to so-called 'flat-foldable' or near-'flat-foldable' vertices; the 'flat-foldability' restriction on angles in a crease pattern leads to dramatic algebraic simplifications in rigidity calculations. As a result, Miura-Ori derivatives are often perfectly foldable, with the stiff sheet between creases (i.e., the 'faces') not bending at all when the creases are folded. However, restricting study to the very special Miura-Ori and the focus on algebraically perfect mechanisms with no face bending might miss a larger universe of near-perfect mechanisms in which face bending or energy cost of actuation can be made arbitrarily small."

The authors are correct in that in terms of sheer number of published papers, most rigid origami studies have focused on flat-foldable and symmetric crease patterns. But very significant and influential work has been done outside of this restricted realm and do not focus on the Miura-ori. Examples are:

T. Tachi's rigid origami simulator work [1] where a perturbation method is used to model the folding of crease patterns that are not necessarily flat-foldable.

Wu and You's quaternion-based rigid origami model [2].

Also, D. A. Huffman's 1977 paper (which the authors cite twice, reference 22 and 24 in their paper) provides everything needed for explicit folding angle relationships in degree-4 vertices that do not necessarily fold flat (although Huffman also discusses the flat-foldable case).

The paper under review does offer a new approach to loop constraint equations that can model arbitrary orders of motion, and they use this to study crease patterns that are approximately rigidly-foldable in a systematic way. This is a very interesting contribution to the field, and it could be viewed as significant enough to warrant publication in a high-impact journal like Nature Communications. I personally do not think it does, but I'd be happy to be overruled.

If the paper is recommended for publication, I strongly suggest that the language used in the paper, such as in the abstract and the above-quoted paragraph, be toned down since there are

widely-read publications that deal with non-flat-foldable rigidity and explicit solutions thereof.

[1] T. Tachi, Simulation of rigid origami, *Origami⁴: Proceedings of the 4th International Meeting of Origami in Science, Mathematics, and Education*, 2009, 175-187.
http://www.tsg.ne.jp/TT/cg/SimulationOfRigidOrigami_tachi_4OSME.pdf

[2] W. Wu and Z. You, Modelling rigid origami with quaternions and dual quaternions, *Proc. of the Royal Society A*, Vol. 466, 2010, 2155-2174
<http://rspa.royalsocietypublishing.org/content/466/2119/2155>

Reviewer #3 (Remarks to the Author):

The review and a recent reference are attached

Reviewer #4 (Remarks to the Author):

This paper describes the analysis on the surface deformation of origami structures which have four vertices with a quadrilateral facet. The authors developed a systematic approach, solving "loop equations", to study non-rigid-foldable structures as well as rigid-foldable structures (e.g., Miura-ori) by measuring the deformation of the quad facet, and this approach is implemented in Matlab codes to investigate 1) statistical relationships between loop equations and the deformation of the quad and 2) statistical relationships between Mountain-Valley crease assignment and foldability. The manuscript describes the detail of their approach, though the authors need to provide more explanations for figures in this paper in order to avoid the confusion for the readers. For example, Fig.1 is mentioned only in page 4 without any explanations, and Fig.3c is not mentioned in the main manuscript. Since self-folding technique has recently gained a large amount of attention due to great potentials for efficient 3D manufacturing technique from surface materials and actuation systems, the authors' work can inspire other researchers to develop practical foldable structures based on origami. For the authors to publish this manuscript, they need to organize the manuscript in a more readable form, especially explanation for figures, so that the readers can follow the flow of the manuscript without confusions.

The authors might also appreciate the detailed comments as below:

1. In page 1, the authors defined "self-folding" based on rigid origami, but is self-folding origami rigid origami? The title of this manuscript contains "self-folding", however the authors discuss non-rigid origami. Also, the authors use "flat-foldability" instead of "rigid-foldability", so it would be helpful to define these terms clearly to avoid the confusion for the readers.

2. For Fig.1, it would be helpful for the readers if the authors explain the details of this figure in main text.

3. In page 3, the authors introduce the angle ϕ to examine the face bending of the quad. The reviewer thinks that it is very important to verify the validity of using ϕ to express the surface bending behavior of their origami structure (e.g., compare with FEM or physical prototype). Can ϕ really represent face bending of actual origami? Figure 2c inset shows the additional crease line whose folding angle is represented by ϕ , and the crease line connects the lower left corner of the quad to the upper right corner. If this diagonal crease line is made connecting the lower right and the upper left, does this affect the authors' calculations significantly? What if two diagonal crease lines are used (i.e., "X" shape crease lines in the quad)? The authors need to elaborate the validity of their approach.

4. For Fig.3c, it would be helpful for the readers if the authors explain more details of this figure in main text. Subsection Also, "Bistability" in page 5, "(Fig.2c)" should be "(Fig.3c)".

5. For Fig.3c, Each curve does not finish at Crease folding of n . Are there any specific reasons? (e.g., singularities)

6. In page 5, the authors mention bistable behavior of origami based on face bending change

along a diagonal crease line in the quad. In their calculations, folding of the other crease lines is not considered. If folding of the other crease lines is considered, do we still obtain the bistable behavior? In particular, if one make a physical prototype of origami based on the authors' work, can the prototype exhibit the bistable behavior?

7. In the last paragraph of Section D. "Discussion", "(as in Fig.2C)" should be "(as in Fig.2c)". (use small letter "c")

8. It would be helpful to mention how one can utilize the authors' work to design self-folding origami devices.

In summary, this manuscript is not recommended for publication in Nature Communications at this moment unless the authors provide answers to questions above. It is recommended that the authors add more explanations to discuss their analysis method and calculation results in the main manuscript.

Reviewer 2 (Remarks to the Author):

The paper under review describes a new approach to modeling rigidly-foldable structures whose lowest-energy motion is parameterized by a single degree of freedom. The approach used consists of characterizing loop equations for the folding angles around a polygon (quadrilaterals, in particular) of the crease pattern in a folded state. The first, second, third, etc. orders of these equations provide conditions for rigid foldability and may be used to describe the space of crease patterns that exhibit rigid-folding behavior. Furthermore, this approach is not reliant on the vertices of the crease pattern being flat-foldable, allowing more generality than is often present in such studies. The paper is an interesting addition to the growing literature on rigid folds. The work appears to be correct and its methodologies sound. But the question is whether this paper's approach is novel and significant enough for publication in Nature Communications. I disagree with the authors on the novel aspects of their approach. For this reason, and this reason alone, I am not recommending the paper for publication in Nature Communications. But the editor(s) may choose to view the other novel aspects of this paper as enough to warrant publication.

We thank the reviewer for his/her detailed review. Indeed, the reviewer's description of the details of our method is accurate. However, our earlier manuscript obscured the novel results obtained through this method. We have now extended our results and rewritten our manuscript to emphasize these results, in particular:

1. Different Mountain-Valley (MV) classes greatly differ in their intrinsic foldability; e.g., $\sim 90\%$ of low energy patterns are of Natural MV type despite constituting only $\sim 6/16$ of all possible MV types.

Despite Mountain-Valley data being one of the most prominent features of classical origami, to our knowledge, this strong relationship between Mountain-Valley choice and foldability was not noticed before.

2. The trade off between number of patterns ('entropy') and foldability ('energy'): Our comprehensive study of all patterns at different energy scales has now resulted in a formula for the total number of crease patterns $e^{S(\bar{E})}$ of area A that have folding energy \bar{E} per unit area,

$$S(\bar{E}) = \frac{A}{2} \log\left(\frac{\bar{E}}{E_0}\right) \quad (1)$$

(See paper for definition of area A , entropy S and energies \bar{E} , E_0 for a quad mesh.)

Such a result is useful to understand the trade-off between folding energy (stiffness) and design freedom. Given a higher actuation energy, how many more patterns become available)? How many patterns have actuation energy higher than a given energy scale of uncontrolled fluctuations, and thus avoid inadvertent folding?

We believe that such methods and results, connecting properties of single crease patterns to ensembles of crease patterns through energy and entropy perspectives are of a new flavor in this inter-disciplinary field. The single pattern results connect to prior mathematical work in origami while ensemble results connect to materials science applications where an error-prone

synthesis method might effectively sample an ensemble of patterns; the ensemble results also connect to earlier work on thermodynamics of thin sheets and crumpling transitions.

We hope our re-written manuscript better communicates these new results and how our work connects with earlier work in these diverse fields.

Therefore I will provide more detailed commentary below. Specifically, the authors state in the third paragraph: "... origami design thus far has mostly been limited to the mountain-valley (MV) pattern implicit in Miura-Ori.[...] might miss a larger universe of near-perfect mechanisms in which face bending or energy cost of actuation can be made arbitrarily small."

The authors are correct in that in terms of sheer number of published papers, most rigid origami studies have focused on flat-foldable and symmetric crease patterns. But very significant and influential work has been done outside of this restricted realm and do not focus on the Miura-ori. Examples are:

T. Tachi's rigid origami simulator work [1] where a perturbation method is used to model the folding of crease patterns that are not necessarily flat-foldable.

Wu and You's quaternion-based rigid origami model [2].

Also, D. A. Huffman's 1977 paper (which the authors cite twice, reference 22 and 24 in their paper) provides everything needed for explicit folding angle relationships in degree-4 vertices that do not necessarily fold flat (although Huffman also discusses the flat-foldable case).

We thank the reviewer for pointing out these references. As the reviewer requested, we have re-written our paper to better situate our work in the context of existing research, added these citations (and other related ones) in the appropriate places, and clarified our claims of novelty.

In particular, we have edited the text quoted above to accurately reflect prior work:

"With the exception of some influential works discussed below[Tachi:2009nb,Huffman:1976,Hull:2002, Tachi:2012,Tachi:2009ub,Wu:2010], [..] origami design has often been limited to the mountain-valley (MV) pattern implicit in Miura-Ori"

We then explicitly discuss these foundational works on non-Miura and non-rigid foldable patterns (cited above) that came before us:

"Important contributions in the study of generic origami patterns include Huffman's work[] on the folding motions of a general n-valent vertex and Tachi's simulation scheme of origami patterns[]. Wu et. al. introduced analytic methods to analyze motions of multi-vertex patterns[], extending Belcastro and Hull's condition for testing rigid-foldability[]. Tachi went beyond rigid foldability for general patterns by establishing design principles for first order foldability[]"

We edited the other text quoted by the reviewer to say:

"Restricting study to the rigid foldable patterns with no face bending .."

where it used to say *"Restricting study to the very special Miura-Ori ..."*

Other edits of similar nature are detailed here. In the abstract, we now say:
"usually only two extreme classes of crease patterns are studied; "
where it used to say *"only two extreme classes of crease patterns are known..."*

Right in the first paragraph of the introduction, we credit prior works on non-Miura non-flat foldable patterns:

" Despite much recent interest in large extended mechanisms[] and some critical contributions towards the same[Tachi:2009nb,Huffman:1976,Hull:2002,Tachi:2012,Tachi:2009ub,Wu:2010], most work has focused on deformations with high symmetry,"

On Page 3 (Results), we now say,
" Similar loop equations for lowest order foldability were derived by Tachi [] earlier."

In Supplementary Note 2, we now say:
"Tachi identified a single loop equation as a necessary and sufficient condition for quads made of flat foldable vertices to also be rigidly foldable with no face bending at all to any order. Tachi later derived a 1-st order loop equation for general non-flat foldable quads."

and connect to Tachi's results explicitly there:
"How do our results for tunable foldability of general crease patterns with arbitrary Mountain-Valley choice relate to Tachi's [] results for rigid foldability of crease patterns with 'flat-foldable' vertices?"

As the reviewer points out and as acknowledged in our current manuscript, the referenced works by Huffman, Tachi, Wu (and others we have now cited, e.g., Belcastro & Hull) were indeed foundational in studying non-Miura non-flat foldable patterns. We have also highlighted another relevant work, Tachi (2010), found during our revision process.

However, these works were either restricted to single vertices, analyzed primarily a few special crease patterns, or provided a numeric exploration tool. All of them, including Tachi's work on 1st order loop equations, ignored the impact of Mountain-Valley data on the foldability of patterns. In contrast, our analytic and systematic approach provides a foundation that, on a practical level allows both design of arbitrarily foldable patterns, and on a theoretical level, shows quantitative relationships between geometry (e.g., Mountain-Valley data), design freedom ('entropy') and foldability ('energy').

We agree with the reviewer the above works need to be cited to set our work in context. We believe our current manuscript better explains how our work builds on the work of others.

The paper under review does offer a new approach to loop constraint equations that can model arbitrary orders of motion, and they use this to study crease patterns that are approximately rigidly-foldable in a systematic way. This is a very interesting contribution to the field, and it could be viewed as significant enough to warrant publication in a high-impact journal like Nature Communications. I personally do not think it does, but I'd be happy to be overruled.

If the paper is recommended for publication, I strongly suggest that the language used in the paper, such as in the abstract and the above-quoted paragraph, be toned down since there are widely-read publications that deal with non-flat-foldable rigidity and explicit solutions thereof. [1] T. Tachi, Simulation of rigid origami, *Origami4: Proceedings of the 4th International Meeting of Origami in Science, Mathematics, and Education*, 2009, 175-187. [2] W. Wu and Z. You, Modelling rigid origami with quaternions and dual quaternions, *Proc. of the Royal Society A*, Vol. 466, 2010, 2155-2174

We thank the reviewer for the comments on the systematic and novel nature of our work and also pointing out the shortcomings in addressing prior work. Our response primarily involves:

- (1) clarifying our claims of novelty and better citations,
- (2) new results and a better presentation of results obscured in the earlier manuscript.

We think our work represents a substantial advance, by studying crease patterns to all orders systematically, uncovering the Mountain-Valley vs foldability relationship and the entropy vs folding energy trade-off. The paradigm of programmed self-folding sheets will continue to find applications at diverse length and energy scales. We thank the reviewer for helping improve our contribution to such an ongoing effort.

Summary of major changes:

1. Finite element simulations of folding using COMSOL. Confirms that our results hold for realistic materials that can bend, stretch, are of finite thickness etc.
2. New experimental realization of a modified self-folding box without symmetry. (Replaced picture in Fig 1C and video in SI)
3. Significant new results (simulations + theory) on the entropy of crease patterns as a function of energy scale.
4. Explicit connections of results to materials science applications by identifying three relevant energy scales, $E_{material}$, $E_{actuation}$, E_{noise} across diverse applications in the literature.
5. Multiple simulations of larger lattices to verify both entropic and energetic results in the paper.
6. Expanded motivation for our study by relating to problems in materials science applications, thermodynamics of crumpling.
7. Revised presentation of our novel result relating Mountain-Valley choice and foldability. New simulations to quantify this Mountain-Valley-foldability relationship using entropy.
8. Added citations to earlier works on non-Miura Ori based patterns. Clarified how our work builds on but goes beyond existing work in the field.

Reviewer 3 (Remarks to the Author):

The paper Self-folding origami at any energy scale investigates possible approximations of rigid-folding necessary conditions in origami. If not exactly satisfied, these non-linear conditions may be satisfied at different order once linearized (loop equations). Numerous simulations performed by the authors shows that the more loop-equations are satisfied, the less bending energy is stored in the considered origami. This suggests that the residual of these equations is a good tool for estimating the foldability of a crease pattern.

Beyond this interesting observation, the actual motivations of the paper are not clear. The authors evoke the universe of self folding without really detailing practical or theoretical consequences. Several other concepts such as entropy or bistability are roughly sketched but not enough developed which leaves the reader expectant.

We thank the reviewer for his/her review and for noting our novel design principle for self-folding origami. We believe the earlier presentation obscured some of our central results and their significance. We have now addressed motivation/significance through revised presentation and new results detailed later. To summarize the highlights:

1. Our work provides a *design principle* to create self-folding origami of any given energy over 9 orders of magnitude. Design tools as a function of a continuum energy scale are crucial for materials applications where intrinsic bending stiffness might vary by as many orders of magnitude. (*revised presentation*)
2. Our energy-entropy relationship quantitatively relates foldability, design freedom and size of patterns; it is relevant for materials applications since it quantifies the trade-off between design freedom and energy scales of actuation and of noisy processes. (*new results*)
3. Mountain-Valley vs foldability: Despite Mountain-Valley data a prominent feature of classical origami, to our knowledge, the strong relationship between Mountain-Valley choice and foldability was not noticed before. (*new results and presentation*)
4. Finite Element Simulations in COMSOL and analytic arguments show when our results might hold for real materials by accounting for stretching, the finite thickness of material, the thickness of creases etc. (*new results*)
5. Our statistical and *ensemble* approach to origami is a novel bridge between the mathematical theory of origami and statistical physics and materials science applications. (*revised presentation*)

The paper is globally difficult to read. The motivations are not well explained in the introduction (only with vague terms and implicit references). The main text is not self sufficient and actually requires to read the supplementary material to grasp the global thinking. There are too many different supplementary materials and figures which gives only a fragmented view of the work. There are many vague terms which make the explanations confusing and some notations are ambiguous in the technical parts. On this point, Section C is particularly confusing with many words in quotes (see also detailed comments).

We recognize that the presentation of results in the earlier manuscript did not do our work justice; we have now revised and extended our results to highlight novel contributions

and their significance, as will be detailed below. In response to the specific comments above:

1. Vague terms: We have followed the reviewer’s detailed comments below and either removed vague terms, or provided a definition and adopted standard terms used in the field.

2. Supplementary materials: have been greatly condensed. We now have only two notes and two figures as supplementary materials.

3. Section C on Entropy has been completely rewritten, highlighting our novel result on the entropy-folding energy trade-off.

4a. Global view: References to Fig 1 and Fig 2 are now used to give a global view of our work - they outline a systematic way of designing a self-folding mode in the folding energy landscape of a sheet. For e.g., on Page 4, we now say,

”as noted in Fig. 1a, the energy required to actuate the designed mode, $E_{\text{designed}} \sim \rho_{\text{face}}^2 \sim \rho_{\text{crease}}^{An+2}$, drops rapidly with the number n of the exactly satisfied loop equations...”

and

”An intuitive way to understand the fine tuning required is to write a consistency loop condition for a fold angle ρ , say that of AD (see Fig.2b), transported around the quad (i.e. forming a closed loop),”

and

”As shown in Fig 2c, the loop equations when solved in sequence provide a controlled and systematic reduction in face bending over 9 orders of magnitude.”

We have also created and photographed a new, more illustrative origami sample for Figure 1, as requested by the reviewer below.

4b. We use Fig 2 to communicate the general idea of a series expansion; we think the algebraic details of the series expansion itself are best presented in a Supplementary Note for a journal like Nature Communications. These details are not crucial to understand that our series expansion systematically reduces the folding energy.

Finally, the authors must consider the work from Tachi (2012) which was very recently improved (Demaine et al., 2016) (a conference two weeks ago, not yet accessible online and attached to the present review). They give a clear geometric interpretation of first and second order loop equations and is extremely close to the approach suggested in the present paper. It is however acknowledged that the results presented here goes beyond these preceding works since the order of the development is greater. Additionally, I strongly suggest to follow the notations and technical passages used in these works since they are more conventional.

We thank the reviewer for bringing this beautiful recent conference proceeding to our attention. Indeed, this is a highly instructive geometric interpretation of part of our method. We were also remiss to not cite Tachi’s 2012 work - much of Tachi’s work was clearly an inspiration to our work, but we overlooked this particular reference. We have now also highlighted another relevant work of Tachi (2010) during our revision process.

For example, in the Introduction, we now say,

”Surprisingly little is known about general self-folding origami patterns that are not exactly rigidly foldable. Important contributions include Huffman’s work[Huffman:1976] on general n -valent vertices and Tachi’s simulation scheme of origami patterns[Tachi:2009ub]. Wu et. al. introduced analytic methods to analyze motions of multi-vertex patterns[Wu:2010],

extending Belcastro and Hull’s condition for testing rigid-foldability[Hull:2002] for non-flat foldable. Tachi went beyond rigid foldability for general patterns by establishing design principles for first order foldability[Tachi:2012, Tachi:2009nb].

and on page 3 we say: *”Similar loop equations for lowest order foldability were derived by Tachi earlier.”*

In Supplementary Note 2, we say, *”Tachi identified a single loop equation as a necessary and sufficient condition for quads made of flat foldable vertices to also be rigidly foldable with no face bending at all to any order. Tachi later derived a 1-st order loop equation for general non-flat foldable quads”*

As the reviewer notes, our work does go significantly further and has more of a materials science and statistical physics flavor; our revised manuscript credits prior work while also highlighting our new results and their significance (a continuum sliding energy scale, impact of Mountain-Valley choice on foldability, energy-entropy trade-off and statistical ensemble results useful for materials science and physics).

Notation: While we have minimized mathematical notation in the main manuscript to target a broader audience, we adopted explicit notation and rigorous definitions in the Supplementary Note on loop equations, in accordance with the reviewer’s later comments. We have also reorganized and cleaned up the Supplementary materials in keeping with the reviewer’s comments below.

Consequently, I suggest to reject with possible resubmission provided the anteriority is addressed and the motivations are clarified.

We are grateful to the reviewer for pointing out the very relevant literature we had missed. We believe the current manuscript situates our work better in the existing stream of research on origami from different fields and highlights the motivation for and significance of our results.

Here are some suggestions of improvement:

Motivate practically the need of going higher order than in Tachi (2012); Demaine et al. (2016).

We have addressed the motivations behind and significance of our work in multiple ways as noted earlier; we expand on some of these points here:

1. **Mountain-Valley - Foldability relationship:** Previous works seem to have missed a remarkable connection between Mountain-Valley choice and foldability. We present these results in our revised Mountain-Valley section and revised Fig. 3. As we note there:

”We also see that 90% of modes softer than $E \sim 10^{-1}$ are accounted for by Natural MV configurations, even though such configurations only account for 6/16 of all Mountain-Valley choices. Most of the remaining 10% are accounted for by Semi-natural configurations (8/16 of all choices). Among stiff modes $E > 10^{-1}$, the situation is reversed and Semi-natural and Unnatural configurations form a majority.”

2. **Need for a continuum energy scale:** Self-folding origami applications range widely in the intrinsic stiffness of the materials used (from Nickel-Titanium alloys to hydrogels) and in the actuation energy available (e.g., in active hinges). Solving only two loop equations may not be enough for a stiff material with low energy actuation hinges (e.g., shape memory polymers). Further, solving a discrete number of loop equations gives only a discrete set of folding energies that might be separated by orders of magnitude for a stiff material; the desired folding energy might lie in this gap.

Hence we need a design principle as a function of a continuous energy scale. For e.g., we now say in the introduction:

Understanding the full space of crease patterns as a function of folding energy is also crucial for self-folding origami applications[Peraza:2014], since applications vary widely in material stiffness and actuation torques (or energies) available. For example, folding a structure made of stiff plates connected by Shape-Memory Polymer hinges [Felton2013-kf,Hawkes2010-qr] that provide low actuation torques might require nearly-rigid foldable patterns; but using Shape-Memory Alloys [Kuribayashi2006-rs] or ionic electroactive polymer[Peraza:2014] hinges that provide higher torques would allow use of less foldable patterns as well. Similarly, we might wish to prevent accidental deployment of a self-folding hydrogel capsule[Shim2012-jj] due to small pH fluctuations, necessitating less foldable patterns.

We quantify these large variations in material stiffness and actuation torques cited above in our revamped Entropy section:

”Self-folding origami applications vary greatly in the energy $E_{material}$ needed to bend an uncreased face to a given angle; e.g., compare a Young’s modulus of $\sim 10^3 Pa$ for hydrogels[] to $\sim 10^7 Pa$ for NiTi alloy in origami stents[]. Similarly, actuation mechanisms for active hinges are diverse, including electric[], optical [], thermal [] and chemical (pH) [] methods. Hence, the actuation energy $E_{actuation}$ provided by active hinges (defined as work done by hinges during folding to 1rad) can vary widely; e.g., compare torques of $\sim 6 \times 10^{-3} Nm$ in 30 mm-long shape-memory polymer hinges [] to $5\times$ or $400\times$ that torque in ionic electroactive polymers or shape-memory alloys respectively[].”

3. **Energy-entropy relationship:** We complement our design tool for creating specific patterns of given folding energy with a statistical analysis of the entropy of all crease patterns of that folding energy. What is the probability that a randomly drawn quad mesh would have folding energy less than a threshold E ? How does this probability change with the threshold E and with the size of the quad mesh A ? Our entropy-energy formula $S = \frac{A}{2} \log E$ answers such questions, as explained in the revamped Entropy section; e.g.,

” the probability of a random pattern (when folded with a fixed MV) being softer than energy E decreases exponentially with mesh size A but only as a power law with E .”

The energy-entropy relationship also describes a **design space vs energy trade-off** in materials science applications; e.g., how many more crease patterns become available for a given increase in available actuation energy? In the revamped Entropy section, after

quantifying the diversity of energy scales across origami applications, we explain the trade-offs:

"Taken together, $E_{actuation}/E_{material}$ can vary greatly across applications. Our energy-entropy relationship shows that the fraction of all patterns suitable for such an application is $\sim (E_{actuation}/E_{material})^{A/2}$ (for large A)."

"Additionally, micron-scale applications might often have a design requirement to prevent inadvertent actuation due to uncontrolled noisy processes of a lower energy scale E_{noise} ; e.g., spontaneous temperature [] or pH fluctuations in hydrogels [] or random mechanical kicks. To avoid inadvertent actuation, the folding energy of patterns must be in the 'Goldilocks' zone between E_{noise} and $E_{actuation}$. The fraction of all patterns in the 'Goldilocks' zone can be computed to be $(E_{actuation}/E_{material})^{A/2} - (E_{noise}/E_{material})^{A/2}$ for large A ."

"In this way, equation (7) provides a basic guideline for how many more patterns become available if the actuation energy $E_{actuation}$ can be raised, say, at the cost of higher power input[] into the hinge or if the energy of uncontrolled processes E_{noise} is lowered."

4. We believe our **ensemble** approach opens up a novel bridge between the mathematical theory of origami and statistical physics and materials science applications. While any ensemble result in statistical physics is necessarily specific to that 'random' ensemble, our ensemble methods and results can be adapted to ensembles relevant to any specific application. E.g., in the introduction, we say:

"Energy scale-dependent origami design and statistical properties of 'typical' patterns are the basic building blocks needed for a physically motivated theory of origami [], relevant to both natural [] and synthetic [] systems."

In the Entropy section, we say:

"While our results were derived for the simplest random ensemble, they can be adapted to other ensembles of patterns relevant to specific applications."

5. Finally, self-folding origami has proven foundational for origami theory; indeed, the reviewer's included PDF reference (Demaine 2016) by leading origami researchers addresses the same problem of higher order foldability. Our work continues this analysis by establishing the finite hierarchy of loop equations that must be solved for a general quad pattern to be rigid foldable. Our manuscript has now been updated with such citations of prior work to situate our paper in this context (detailed earlier).

Go into more details about the notion of entropy and its possible consequence regarding order/disorder. Take position with respect to preceding research regarding crumpled paper (fractal organization, competition between membrane/bending energy etc.).

Indeed, we have expanded our treatment of entropy now. In order to expand these discussions and understand the relationship of our work to the areas suggested by the reviewer, we collaborated with Prof. Thomas Witten, an expert in this area of disordered thin sheets

and crumpling. He is now a co-author of this paper.

To summarize changes made:

1. We have revamped our discussion of entropy with new simulations and analysis. Our comprehensive study of all patterns at different energy scales has now resulted in a formula for the total number of crease patterns $e^{S(\bar{E})}$ of area A that have folding energy \bar{E} per unit area,

$$S(\bar{E}) = \frac{A}{2} \log\left(\frac{\bar{E}}{E_0}\right) \quad (2)$$

(See paper for definition of area A , entropy S and energies \bar{E} , E_0 for a quad mesh.)

As described earlier, such a result is useful to understand the trade-off between folding energy (stiffness) and design freedom.

This entropy-energy relationship also connects to earlier attempts to understand the thermodynamics of crumpling transitions. Those earlier entropy-energy relationships were derived in highly simplified ‘lattice’ models, e.g., that creases could only be at 60 degrees to each other or discretized energy into a few values (e.g., zero and non-zero energy). We have cited and connected to those works in our revised entropy section. However, exploring the thermodynamics of quad meshes in greater detail is beyond the scope of this origami paper.

2. We have added a discussion on the competition between bending and stretching (or membrane) energy and how bending is descriptive for the total energy in the limit of thin sheets. We checked these theoretical arguments against Finite Element Simulations using the COMSOL software package. These additions are detailed later in this response.

Make a video of example in Fig 1a).

1. In response to the reviewer’s later request, we synthesized a new self-folding sheet for Fig 1c. We have made a new video and still image of that new origami structure and submitted it.

2. However, we found that a video of the generic unfoladble sheet shown in Fig 1a is not very insightful; Fig 1a is a schematic to illustrate the central approach of the paper; the figure contrasts an unfoldable sheet that solves no loop equations (whose energy scales as ρ^2) with a foldable sheet that solves n loop equations (energy scales as ρ^{4n+2}). The generic unfoldable sheet shown in Fig 1a cannot fold very much anyway and it is hard to see such changes visually in a video.

Simulate a larger quad mesh.

Indeed, a salient feature of our loop equations is that they are applicable to a quad mesh of any size.

1. We studied a 3×3 quad mesh in detail to show that our loop equations, applied to individual quads of a quad mesh, accurately predicts the folding behavior of the quad mesh as a whole. The results are shown in Supplementary Figure 1; in the caption we say,

“face folding of quads that are part of a large pattern closely approximates the behavior of each quad when ‘cut out’ and folded in isolation. Thus the loop equations, applied quad by quad to tune individual foldability, can be used to design a large pattern of desired foldability.”

2. The expanded entropy section is dedicated to large meshes; in particular, we generate and simulated the folding of over 10^5 quad meshes of varying size up to 3×2 . We were able to show that the folding stiffness of these quad meshes obeys a remarkable distribution, summarized by our entropy-energy trade-off $S \sim (A/2) \log E$ noted earlier. The data is presented in Fig. 4 of the main paper and the entire section is dedicated to a discussion of large meshes.

Detailed comments:

1. The term perfectly-foldable is not clearly defined. Is it for rigid-foldable?

Thank you for pointing out this imprecise language. In the current manuscript, we have changed "perfectly-" to "rigidly-" to conform to conventions of the field. For example, on Page 1, we now say

"Miura-Ori derivatives are often rigidly foldable"
instead of *"perfectly foldable."*

2. The word universe is vague. Please use more specific denominations (space, configurations,etc.).

Again, we have revised our paper to avoid such imprecise language. For e.g., on the first paragraph, we now say *"space of designed disordered deformations"* rather than *"universe of designed but disordered deformations."*

3. The suggested videos are irrelevant and not illustrative (remove also the audio!). The corresponding crease patterns in Fig1 are actually rigid-foldable (unless some shaking of the coordinates was introduced... In such case, it should be explicit and motivated). See the work from Kling (2005)

1. As requested by the reviewer, to replace Fig 1c, we created a new asymmetric pattern with no symmetry reasons to be rigid-foldable; yet the pattern is quite foldable because it solves two loop equations exactly. We created a new video and replaced Fig 1c with an image of this non-rigid foldable pattern.

2. Our earlier manuscript did not make it clear that the pattern in Fig 1d is *not* rigid foldable - there are no symmetries that make it so. In fact, by construction, it only solves one loop equation. We now say, e.g., in the caption to Fig 1,

"These patterns solve exactly only one (d) or two (c) loop equations."

3. While the pattern in Fig 1d might resemble some of Kling's patterns superficially, the patterns are quite different. Unlike Fig 1c and Fig 1d, Kling's patterns are rigid foldable due to symmetries.

4. We have now removed the audio in all videos, as requested.

We believe videos of non-symmetric patterns are illustrative since they show how such designed patterns are foldable with minimal control (i.e., are 'self-folding'). Further, the specific patterns in Fig 1c and 1d are useful in drawing a contrast to commonly studied patterns since Fig 1c is very far from flat-foldable and Fig 1d's Mountain-Valley pattern is very different from Miura-Ori.

4. What is the use of full transfer formulas since they are linearized in the end?

We thank the reviewer for pointing out the need to clarify the use of higher order terms in the transfer functions of equations (3-4). Linear terms define the first loop equation $\Pi R = 1$ and were useful to understand the Mountain-Valley-foldability relationship. Higher order terms of the full transfer function are crucial in determining the higher order loop equations.

We changed the text in page 3 to reflect the importance of nonlinear terms:

"We can write a series of loop equations, defined term by term using the expansion of the transfer function of equation (4)."

5. Only bending energy is considered as a consequence of a motion which is not exactly rigid foldable. However, it is expected that also membrane energy may be involved in more general configurations than the example in the paper. Any comment on that?

We thank the reviewer for pointing out that the energetic cost of deformed membranes is not only determined by bending, but by stretching as well. A thin membrane, forced to deform due to held boundaries, will tend to optimize its deformation energy by creating a new well defined crease (see Lobkovsky et. al., Science 270, 5241 (1995)). In this optimized configuration (that is similar to our own augmented quad) bending and stretching energies are comparable. In fact, in the thin sheet limit, a virial argument shows that bending energy is about 5 times larger than the stretching energy. Similarly, stretching strain is much smaller than bending strain in low energy configurations for thin sheets. Taken together, both energy and geometry can be approximated by consider bending alone in the limit of thin sheets.

1. We have updated the manuscript with these notes; e.g. on page 3,

"We note that stretching energy in thin sheets scales the same way with ρ_{face} as bending energy due to a virial theorem[Witten1,Witten3], but is expected to be considerably smaller. Further, for thin sheets, bending strain is much larger than stretching strain in low energy configurations. Hence, in the following, we model both the energy and geometry by considering only face bending. We later check the validity of this thin sheet approximation using finite element simulations in COMSOL (see Fig.5 and Supplementary Figure S.1.)"

"Thickness in real application varies, e.g., 0.05 mm thick NiTi sheets of width 50mm for stents to 1 um thick GaAs sheets of width 100 um for optically actuated mirrors."

2. We performed a Finite Element Simulation in COMSOL to illustrate the effect of sheet thickness; the results are seen in Fig.5 and in Supplementary Figure 1. As we note in the final section on folding curves:

"To visualize what face bending might look like in a real material without an added diagonal, we show results of a finite element simulation in COMSOL in Fig.5b of two select modes from Fig.5a. Unlike our simplified model, this simulation accounts for stretching, bending of all 9 faces, finite thickness of the material and finite width of creases. While our simple diagonal crease model cannot capture the precise folding energies seen in the COMSOL simulation, we see that the bending stress is localized to a furrow along the diagonal, a result expected for thin sheets[] (see Supplementary Figure 1 for more analysis and simulations)."

6. Just before Entropy of solutions: What is the distance from flat foldability. Please, define or give reference.

7. The last sentence before section C is definitely unclear: Alternatively, we could program two completely unrelated folding modes to first order in one quad mesh by imposing the first loop equation for both desired modes.

The confusing expression and sentence in question no longer appear - as noted earlier, we have overhauled the section on entropy.

8. Sup Note 1. End of first section. Please explain why: The larger value of R should be used when ρ_α and ρ_γ have opposite signs (i.e., are of opposite Mountain-Valley state).

We thank the reviewer for his/her close reading!

This issue concerns a bifurcation found for a 4-vertex; a 4-vertex has two distinct non-linear modes that meet at the flat state. Such a bifurcation is briefly noted in the paper included by the reviewer (Demaine 2016) and was the subject of Waitukaitis et al (2014) which we had cited in our main paper. The quadratic equation for R gives two solutions corresponding to each of these branches and should be matched up based on Mountain-Valley pattern. We have now added a citation to Waitukaitis et al in the Supplementary Note and explained:

"One intuitive explanation is that if the angles at the vertex were all 90 degrees, R would be infinite if ρ_α, ρ_γ have opposite signs."

9. Sup Note 1, section 2: Please write explicitly z , define c_i (is it related to $c_{\alpha\beta}$?) and define R_i (is it related to $R_{\alpha\beta}$?)

We have now edited our Supplementary Note to explicitly define our terms. For example, we now say,

"We may rewrite this equation as $z_V \equiv \rho_i - T_V(\rho_{i+1}) = 0$ " and "we define $\mathbf{z} \equiv \sum_V z_V = 0$."

We have defined c_i, R_i in a similar manner.

References:

T. Tachi, Design of infinitesimally and finitely flexible origami based on reciprocal figures, J. Geom.Graph. 16 (2012) 223234.

E. D. Demaine, M. L. Demaine, D. A. Human, T. C. Hull, D. Koschitz, T. Tachi, Zero-Area ReciprocalDiagram of Origami, in: K. Kawaguchi, M. Ohsaki, T. Takeuchi (Eds.), Proc. IASS Annu. Symp.,Tokyo.

D. H. Kling, Patterning technology for folded sheet structures, 2005

In summary, we thank the reviewer for the close reading of our manuscript, including the supplementary information.

We believe we have addressed all the concerns raised by the reviewer through a combination of (a) new results, (b) editing the text for clarity, (c) more specific and concrete motivation and (d) relevant citations. We believe that our manuscript is much improved as a result and will contribute significantly to the conversation in this rapidly growing field.

Summary of major changes:

1. Finite element simulations of folding using COMSOL. Confirms that our results hold for realistic materials that can bend, stretch, are of finite thickness etc.
2. New experimental realization of a modified self-folding box without symmetry. (Replaced picture in Fig 1C and video in SI)
3. Significant new results (simulations + theory) on the entropy of crease patterns as a function of energy scale.
4. Explicit connections of results to materials science applications by identifying three relevant energy scales, $E_{material}$, $E_{actuation}$, E_{noise} across diverse applications in the literature.
5. Multiple simulations of larger lattices to verify both entropic and energetic results in the paper.
6. Expanded motivation for our study by relating to problems in materials science applications, thermodynamics of crumpling.
7. Revised presentation of our novel result relating Mountain-Valley choice and foldability. New simulations to quantify this Mountain-Valley-foldability relationship using entropy.
8. Added citations to earlier works on non-Miura Ori based patterns. Clarified how our work builds on but goes beyond existing work in the field.

Reviewer 4 (Remarks to the Author):

This paper describes the analysis on the surface deformation of origami structures which have four vertices with a quadrilateral facet. The authors developed a systematic approach, solving loop equations, to study non-rigid-foldable structures as well as rigid-foldable structures (e.g., Miura-ori) by measuring the deformation of the quad facet, and this approach is implemented in Matlab codes to investigate 1) statistical relationships between loop equations and the deformation of the quad and 2) statistical relationships between Mountain-Valley crease assignment and foldability.

We thank the reviewer for his/her close reading of our paper. This is indeed a good summary of our method and results.

We are particularly excited by our statistical and energy-dependent results that connect mathematical approaches to origami to materials and physical applications of self-folding origami.

The manuscript describes the detail of their approach, though the authors need to provide more explanations for figures in this paper in order to avoid the confusion for the readers. For example, Fig.1 is mentioned only in page 4 without any explanations, and Fig.3c is not mentioned in the main manuscript. Since self-folding technique has recently gained a large amount of attention due to great potentials for efficient 3D manufacturing technique from surface materials and actuation systems, the authors work can inspire other researchers to develop practical foldable structures based on origami.

We thank the reviewer for the positive comments;

1. Our earlier text did not exploit and explain our figures appropriately. We also had some typos (e.g., Fig 3c mentioned above was mis-cited as Fig 2c). We have now extensively edited explanations to all figures as suggested by the reviewer. E.g., references to Fig 1 and Fig 2 are now used to give a global view of our work. All these changes are detailed in response to comment (2) by the reviewer below.

2. Our revised manuscript puts even more emphasis how our energy-entropy results connect to practical applications, e.g., helping understand the trade-offs in using different active actuation methods in the hinges and in using surface materials of different elastic modulus; see our detailed response to reviewer's comment (7) below.

For the authors to publish this manuscript, they need to organize the manuscript in a more readable form, especially explanation for figures, so that the readers can follow the flow of the manuscript without confusions.

We appreciate the reviewer's point; as detailed below, we have re-organized the flow around our figures, eliminated less significant details and ambiguous terminology, and explicitly connected the mathematical end of our work to statistical physics and materials science applications.

The authors might also appreciate the detailed comments as below:

1. In page 1, the authors defined self-folding based on rigid origami, but is self-folding origami rigid origami? The title of this manuscript contains self-folding, however the authors discuss non-rigid origami. Also, the authors use flat-foldability instead of rigid-foldability, so it would be helpful to define these terms clearly to avoid the confusion for the readers.

We thank the referee for pointing out this imprecise language.

1. We have now defined our terms; e.g., in the introduction, we clarify ‘self-folding’:

”If creases are placed at just the correct angles relative to each other, the sheet as a whole has exactly one allowed deformation in which all the creases fold at the same time. Such sheets can be described as self-folding because the allowed mode will be actuated by almost any applied force.”

2. We then define rigid foldability in the introduction as *”foldable at precisely zero energy cost”*.

3. Flat foldability is an existing term in origami; in the introduction, we have now defined flat foldable origami in the as *”patterns in which all creases fold to angle π simultaneously.”* (Note that flat foldability only concerns the existence of a fully folded state while rigid foldability and self-folding concerns the energy during the entire folding process.)

2. For Fig.1, it would be helpful for the readers if the authors explain the details of this figure in main text.

Indeed, we have corrected such oversights for Fig 1 and other figures as well. We detail such these changes here:

We use Fig 1 and Fig 2 to give a global overview of our work. For example, early in the introduction, we now say:

”Figure 1a illustrates this idea: a general origami pattern might have several folding motions, but a self-folding pattern will have a unique extended motion that requires less energy than all others.”

We then use Fig 1b to explain past work on rigid foldable patterns:

”Rigidly-foldable crease patterns are often periodic structures made of repeating units, such as Miura Ori (Fig.1b) and its derivatives []. With the exception of some influential works discussed below[] origami design has often been limited to the mountain-valley (MV) pattern implicit in Miura-Ori seen in Fig.1b. []”

After presenting the motivation for energy-dependent design, we say :

”examples of how our results can be used to produce large quadrilateral meshes with non-Miura geometric and mechanical properties (Fig. 1c,d).”

Note that we also created and photographed a new origami sample with less symmetry to replace the older Figure 1c, in response to another reviewer.

In the section on loop equations, we now say,

”as noted in Fig. 1a, the energy required to actuate the designed mode, $E_{\text{designed}} \sim \rho_{\text{face}}^2 \sim \rho_{\text{crease}}^{4n+2}$, drops rapidly with the number n of the exactly satisfied loop equations...”

We use Figure 2 to explain the loop equations further in the same section:

”An intuitive way to understand the fine tuning required is to write a consistency loop condition for a fold angle ρ , say that of AD (see Fig.2b), transported around the quad (i.e. forming a closed loop),”

”As shown in Fig 2c, the loop equations when solved in sequence provide a controlled and systematic reduction in face bending over 9 orders of magnitude.”

As requested by the reviewer below in comment (4), we have overhauled our references to the figure on folding curves and bistability (old Fig 3, new Fig 5). We now explain on Page 7,

”In Fig.5a, we show face bending as a function of crease folding for four different folding modes. We see that the loop equation residue $|\Pi R - 1|$ is a good predictor of face bending up to crease folding of $\sim \pi/2$ radians. For even larger folding, strongly non-linear effects kick in; the initially-soft red mode become stiffer than others rapidly while other modes even show non-monotonic behavior. [...]”

We then explain the finite element simulation in Fig 5b:

”To visualize what face bending stresses might look like in a real material without a diagonal crease, we show results of a finite element simulation in COMSOL in Fig.5b of two select modes from Fig.5a. Unlike our simplified model, this simulation accounts for stretching, bending of all 9 faces, finite thickness of the material and finite width of creases. [...] we see that the bending stress is localized to a furrow along the diagonal, a result expected for thin sheets[] (see Supplementary Figure 1 for more analysis and simulations). Further, when folded in COMSOL with a small but fixed folding force , [...]”

For more on Fig 5, see response to comments (3) and (6) below.

3. In page 3, the authors introduce the angle ρ_{face} to examine the face bending of the quad. The reviewer thinks that it is very important to verify the validity of using ρ_{face} to express the surface bending behavior of their origami structure (e.g., compare with FEM or physical prototype). Can ρ_{face} really represent face bending of actual origami? Figure 2c inset shows the additional crease line whose folding angle is represented by ρ_{face} , and the crease line connects the lower left corner of the quad to the upper right corner. If this diagonal crease line is made connecting the lower right and the upper left, does this affect the authors calculations significantly? What if two diagonal crease lines are used (i.e., X shape crease lines in the quad)? The authors need to elaborate the validity of their approach.

6. In page 5, the authors mention bistable behavior of origami based on face bending change along a diagonal crease line in the quad. In their calculations, folding of the

other crease lines is not considered. If folding of the other crease lines is considered, do we still obtain the bistable behavior? In particular, if one make a physical prototype of origami based on the authors work, can the prototype exhibit the bistable behavior?

We thank the reviewer for raising these two related and important points; We choose to address both points here since they both concern the validity of our face bending model for real materials. We had indeed studied this issue extensively but did not include the results because of paper length considerations.

Our conclusion is that any means of augmenting a simple quad to add a single degree of freedom as a measure of ‘unfoldability’ does not qualitatively affect the observed folding modes and folding energies. We now support this conclusion through theory, more simulations of our models with different diagonals and finite element simulations in COMSOL.

1. We have dedicated a 6-panel Supplementary Figure 1 to this issue of model validity. In particular, we simulated patterns with an X shaped diagonal, only the ‘left’ diagonal and only the ‘right’ diagonal; as seen in Supplementary Figure 1(a), all three methods give the same qualitative face bending curves. The caption reads in part:

”Whether we augment with the usual face diagonal, the other face diagonal or an X shape double diagonal, face folding along different folding modes reported in Fig.4 of the paper remains qualitatively the same. In fact, any augmentation that adds one degree of freedom to the simple quad does not change its folding modes.”

2. To account for bending of all 9 faces, stretching, finite width and thickness of creases etc, we carried out a Finite Element Simulation in COMSOL that is presented in Fig. 5b. As we say in the text there:

”Unlike our simplified model, this simulation accounts for stretching, bending of all 9 faces, finite thickness of the material and finite width of creases. While our simple diagonal crease model cannot capture the precise folding energies seen in the COMSOL simulation, we see that the bending stress is localized to a furrow along the diagonal, a result expected for thin sheets[] (see Supplementary Figure 1 for more analysis and simulations). Further, when folded in COMSOL with a small but fixed folding force (note: not to fixed angle), the red mode shows higher stresses, implying that it is softer than the green mode, in agreement with our face bending model in Fig.5a. ”

3. We also ran distinct finite element simulations (in COMSOL Multiphysics) of the center plates of different thicknesses to show that our face diagonal model is a good approximation of the stressed geometry for thin plates; see Supplementary Figure 1c for which the caption reads:

”Finite element simulations of the center plate show that face diagonals are good approximations for the geometry of thin origami patterns. We applied boundary conditions corresponding to an origami folding mode; we found that the elastic energy (and strain) is increasingly localized to a diagonal furrow as the thickness t of the plate is decreased relative to lateral dimensions L . (Here, Young’s modulus $Y = 3 \text{ GPa}$ (material = PVC), $L \sim 10\text{cm}$. Simulations using COMSOL.)”

4. We also explain the theoretical limits of validity of our thin sheet model in the main paper;

"We note that stretching energy in thin sheets scales the same way with ρ_{face} as bending energy due to a 'virial theorem'[], but is expected to be considerably smaller []. Further, for thin sheets, bending strain is much larger than stretching strain in low energy configurations []. Hence, in the following, we model both the energy and geometry by considering only face bending. We later check the validity of this thin sheet approximation using finite element simulations in COMSOL (see Fig.5 and Supplementary Figure S.1.). Thickness in real application varies, e.g., 0.05 mm thick NiTi sheets of width 50mm for stents [] to 1 um thick GaAs sheets of width 100 um [] for optically actuated mirrors. "

5. We also investigated the reviewer's issue for diagonals on a larger lattice; see Supplementary Figure 1b.

"We see that the face folding of quads that are part of a large pattern closely approximates the behavior of each quad when 'cut out' and folded in isolation. Thus the loop equations, applied quad by quad to tune individual foldability, can be used to design a large pattern of desired foldability."

6. Bistability: Based on all the simulations and theoretical arguments outlined above, we have good reason to believe that our face bending model is sufficiently accurate to predict bistability for real materials. An earlier experiment on hydrogels did see bistability where presumably all faces can bend (albeit for a special symmetric quad with 90° angles with special MV data). We have noted this on Page 7 as,

"In fact, such non-monotonic bistable behavior has been seen before in experiments on highly symmetric flat-foldable patterns[Silverberg:2015gb,Dudte:2016db]."

In the current manuscript, we have reduced the focus on the bistability and instead focus more generally on the limits of the validity of our modeling for very large folding angles, when the plates have thickness, finite crease width, and stretching, bending of other faces etc. As we say in that revamped final subsection on Page 7:

"In this work, face bending was measured by augmenting the quad with a diagonal crease and then setting crease folding to one representative non-linear angle (~ 1 radian) in equation (6). How does face bending behave as a function of crease folding? How might these face bending stresses look like in a real material without an added diagonal crease? "

In carrying out the above work (and also for the new work on the entropy-energy relationship), we sought help from Prof. Thomas Witten, an expert on thin sheets and crumpling. He has now been added as a co-author.

4. For Fig.3c, it would be helpful for the readers if the authors explain more details of this figure in main text. Subsection Also, Bistability in page 5, (Fig.2c) should be (Fig.3c).

We thank the reviewer for pointing out the lack of clarity in figure 3c, as well as the mistake in referencing it.

We have now edited the text references this figure (which is now Fig 5a) as well, as detailed in response to comment (2) and (3) of the reviewer above.

5. For Fig.3c, Each curve does not finish at Crease folding of . Are there any specific reasons? (e.g., singularities)

We thank the reviewer for the close reading! We stopped the simulation of pattern folding for creases angles near $\sim 99\%\pi$ because of numeric issues with integration; for one of the modes, this numeric issue set in earlier $\sim 90\%\pi$ but does not represent a real singularity.

1. The new Supplementary Figure 1 shows that this numeric issue is somewhat reduced when using the other face diagonal.

2. We have added a note about our folding procedure in the Supplementary methods section:

”To improve numerical stability, we start in the folded state where the maximum crease angle is 1rad. We then fold backwards toward the flat states at intervals of 0.05rad, and forward towards π using the same interval. We note that patterns are folded until the maximum crease angle reaches 3.1 rad. We avoid folding very close to angle π , as this angle value indicates a collision of faces, after which the system leaves the domain of validity of our analysis.”

7. In the last paragraph of Section D. Discussion, (as in Fig.2C) should be (as in Fig.2c). (use small letter c).

We have now fixed this in our updated manuscript.

8. It would be helpful to mention how one can utilize the authors work to design self-folding origami devices.

We have now added substantially to this discussion:

1. **Energy-dependent design:** Self-folding origami applications range widely in the intrinsic stiffness of the materials used (from Nickel-Titanium alloys to hydrogels) and in the actuation energy available (e.g., in active hinges). Our energy-dependent design principle allows you to create patterns appropriate for any particular application by solving the appropriate number of loop equations. For e.g., we now say in the introduction:

Understanding the full space of crease patterns as a function of folding energy is also crucial for self-folding origami applications[Peraza:2014], since applications vary widely in material stiffness and actuation torques (or energies) available. For example, folding a structure made of stiff plates connected by Shape-Memory Polymer hinges [Felton2013-kf,Hawkes2010-qr] that provide low actuation torques might require nearly-rigid foldable patterns; but using Shape-Memory Alloys [Kuribayashi2006-rs] or ionic electroactive polymer[Peraza:2014] hinges that provide higher torques would allow use of less foldable patterns as well. Similarly, we might wish to prevent accidental deployment of a self-folding hydrogel capsule[Shim2012-jj] due to small pH fluctuations, necessitating less foldable patterns.

Thus our loop-equation based design principle provides a way to pick patterns appropriate for a given material stiffness and given actuation mechanism. e.g., we note in the loop equation section:

"... Equation (5) thus provides a simple design principle for exploring the crease patterns at any chosen folding energy scale over many orders of magnitude; one simply solves the hierarchy of loop equations to the extent needed. "

2. Through **finite element simulations** in COMSOL, we have shown to what extent our idealized model holds when including complications in real applications such as stretching, finite width of creases and the thickness of the sheet. In particular, we show that our design principles are most useful for thin sheets. (detailed earlier)

3. **Tolerance to errors and Mountain-Valley data:** We present a remarkable quantitative connection between Mountain-Valley choice and foldability that previous works have missed, despite the prominence of Mountain-Valley data in origami.

The statistical nature of these results about typical patterns in an *ensemble* makes them relevant to physical and materials applications where a manufacturing process might have a known error distribution in, say, the placement of vertices or creases. As we say in our Mountain-Valley section:

"We also see that 90% of modes softer than $E \sim 10^{-1}$ are accounted for by Natural MV configurations, even though such configurations only account for 6/16 of all Mountain-Valley choices. Most of the remaining 10% are accounted for by Semi-natural configurations (8/16 of all choices). Among stiff modes $E > 10^{-1}$, the situation is reversed and Semi-natural and Unnatural configurations form a majority."

"Such widely varying entropy of MV classes suggests important lessons in design; Natural configurations can be expected to be more forgiving of error in laying out creases while Unnatural configurations need to be highly fine-tuned to be foldable."

4. **Trade-off between available patterns and actuation energy:**

Our energy-entropy formula $S = \frac{A}{2} \log E$ explains the consequences of the choice of actuation mechanism and surface material on the number of crease patterns available. We begin with idealized questions in the revamped Entropy section;

e.g., *"the probability of a random pattern (when folded with a fixed MV) being softer than energy E decreases exponentially with mesh size A but only as a power law with E ."*

In the same section, we then explain how our energy-entropy relationship gives the trade-off between patterns available and energy scales:

"... an entropy-energy relationship quantifying the number of crease patterns of giving folding energy. This relationship describes how many more crease patterns become available for a given increase in available actuation energy, say, in active hinges."

"Self-folding origami applications vary greatly in the energy E_{material} needed to bend an uncreased face to a given angle; e.g., compare a Young's modulus of $\sim 10^3$ Pa for hydrogels[]

to $\sim 10^7$ Pa for NiTi alloy in origami stents[]]. Similarly, actuation mechanisms for active hinges are diverse, including electric[], optical [], thermal [] and chemical (pH) [] methods. Hence, the actuation energy $E_{actuation}$ provided by active hinges (defined as work done by hinges during folding to 1rad) can vary widely; e.g., compare torques of $\sim 6 \times 10^{-3}$ Nm in 30 mm-long shape-memory polymer hinges [] to $5\times$ or $400\times$ that torque in ionic electroactive polymers or shape-memory alloys respectively[]].”

Such considerations are important in deciding the choice of actuation mechanism and surface material through two energy scales $E_{actuation}$, $E_{material}$, as we explain:

”Taken together, $E_{actuation}/E_{material}$ can vary greatly across applications. Our energy-entropy relationship shows that the fraction of all patterns suitable for such an application is $\sim (E_{actuation}/E_{material})^{A/2}$ (for large A). ”

Thus our energy-entropy relationship describes a design space vs energy trade-off in materials science applications; e.g., how many more crease patterns become available for a given increase in available actuation energy?

5. Accidental deployment In the same context, we explain that we can prevent accidental deployment of self-folding origami using our design principles:

”Additionally, micron-scale applications might often have a design requirement to prevent inadvertent actuation due to uncontrolled noisy processes of a lower energy scale E_{noise} ; e.g., spontaneous temperature [] or pH fluctuations in hydrogels [] or random mechanical kicks. To avoid inadvertent actuation, the folding energy of patterns must be in the ‘Goldilocks’ zone between E_{noise} and $E_{actuation}$. The fraction of all patterns in the ‘Goldilocks’ zone can be computed to be $(E_{actuation}/E_{material})^{A/2} - (E_{noise}/E_{material})^{A/2}$ for large A .”

We summarize,

”Equation (7) thus provides a basic guideline for how many more patterns become available if the actuation energy $E_{actuation}$ can be raised, say, at the cost of higher power input[] into the hinge or if the energy of uncontrolled processes E_{noise} is lowered.”

In summary, this manuscript is not recommended for publication in Nature Communications at this moment unless the authors provide answers to questions above. It is recommended that the authors add more explanations to discuss their analysis method and calculation results in the main manuscript.

We thank the reviewer for the detailed account of our manuscript and the useful commentary. Our updated manuscript is considerably improved and to the our best judgment not only clarifies our results, but present a new and interesting take on the characteristics of quad origami mesh design space. We quantify the multitude of available patterns and show how one trades stiffness for design freedom. We also illustrate how these findings may help create origami patterns with a ”Goldilocks” folding energy scale, in principle widely separated from material bending and noise energy scales.

Summary of major changes:

1. Finite element simulations of folding using COMSOL. Confirms that our results hold for realistic materials that can bend, stretch, are of finite thickness etc.
2. New experimental realization of a modified self-folding box without symmetry. (Replaced picture in Fig 1C and video in SI)
3. Significant new results (simulations + theory) on the entropy of crease patterns as a function of energy scale.
4. Explicit connections of results to materials science applications by identifying three relevant energy scales, $E_{material}$, $E_{actuation}$, E_{noise} across diverse applications in the literature.
5. Multiple simulations of larger lattices to verify both entropic and energetic results in the paper.
6. Expanded motivation for our study by relating to problems in materials science applications, thermodynamics of crumpling.
7. Revised presentation of our novel result relating Mountain-Valley choice and foldability. New simulations to quantify this Mountain-Valley-foldability relationship using entropy.
8. Added citations to earlier works on non-Miura Ori based patterns. Clarified how our work builds on but goes beyond existing work in the field.

REVIEWERS' COMMENTS:

Reviewer #2 (Remarks to the Author):

The authors have addressed all my comments and concerns from the earlier draft. I now recommend the paper for publication in Nature Communications.

Reviewer #3 (Remarks to the Author):

Please note that I cannot give an expert opinion on the statistical physics approach.

The main text of the paper was indeed improved and the section regarding entropy presents new and potentially interesting results. However, some important issues need to be addressed. Furthermore, technical information is still too fragmented in the supplementary material and there are still many imprecisions and vague justifications.

Hence I suggest a major revision of the paper.

General comments and questions:

In the section « Entropy energy relationship... », the authors put the stress on the actuation energy of the self-folding system they investigate. However, the influence of the folds own bending energy (which is must be present compared to the vast range of energies under consideration) is never discussed. One expects that this "folds bending" energy sets a lower bound to the total self folding energy which may not be resolved with higher order loop equations. Does this change the analysis (regarding the "Goldilocks" energy range for instance)?

In relation to the previous question, are there indeed real physical systems which actually comply with the assumptions made in the paper (rigid folding, no bending energy along the fold, disordered patterns, actuation energy)? (Note that practical systems which are designed to be self-foldable would not be disordered.) If yes, this does not seem clear in the paper (many references are given but are often a bit far from the exact framework under consideration). Describing precisely such systems would give more importance to the entropy relation the authors derived and may suggest its confrontation to experimental evidence.

Fig1-c and d are still too close to some classical rigid foldable creases patterns. Roughly speaking, showing only that « node shaking » still gives almost rigid-foldable patterns is not very surprising and mitigates the interest of solving higher-order loop equations. It would give much more strength to the paper if the authors exhibit a kind of random pattern which presents the attracting property of being quasi rigid foldable. The corresponding video is the one to be in the supplementary materials. Possibly this pattern may present several folding paths with different energy scales which would also be present in the video.

The discussion about the number of independent loop equations compared to the number of design parameters is very interesting and should be justified/investigated further (currently there are only assertions and empirical testing in the sup. note)

The use of « we » is excessive in the whole manuscript (especially in the supplementary notes) and should be made only when it is a conceptual choice of the authors. Otherwise the passive form is perfectly suitable (typically when talking about simple facts or direct consequences). This excessive use of « we » gives the feeling that the authors are giving themselves credits for already known results:

« our loop equations », low order loop equations are already known.

« we find that the first loop equation » : this is already known
« our strategy » : the expansion of the non-linear loop equation (yet tedious) is actually classical in perturbation theory.
etc.
Being mixed with already known results this does not shed enough the light on what is actually new in the paper (going higher order, entropy etc.)

Detailed Comments:

_Videos are missing in the material transmitted to the reviewer.
_Main Text. p3 first sentence; « edges are too soft »: why using the fuzzy word « soft » whereas the text in the parenthesis is simple and accurate?
_Main Text, equation 3: it is conventional that three lines means « definition ». « = » is preferred.
_Main Text, three § after equation 5: « augmented quads » should be properly defined.
_Main Text, p5 §3: The word « naturalness » sounds awkward. Refer simply to the « class » of a MV sequence.
Main Text, p5 §4: the normalization with ρ{crease}^2 is missing in the definition of $\log E$.
_Fig 3b: name of the horizontal axis: the square seems to apply to the whole \ln function which is ambiguous.
_Main Text p6 §3: what is « plaquette »?
_Main Text p6 4§ after equation 7: the authors refer to the Sup Note 2, but the evoked details do not seem present in the note.
_Sup. Note, page 3, §1: replace « enveloping » by « adjacent »
_Sup. Note, Figure page 3: the symbols are too small
_Sup. Note, First Figure page 4: the symbols are too small
_Sup. Note, page 4, §1, last sentence. The correction mentioned in the authors' answer to me is not present. Furthermore, the sentence itself is still difficult to understand: « R » is corresponding to which creases and which case on the figure (which is barely readable)? Finally, this is where the choice of MV comes to place: this should be linked with the MV class in the main paper and explained to which extent all possible choices were investigated in the paper.
_Sup. Note, page 4, « Strategy »: Notations are still confusing in spite of previous request. They may be clear for the author and kind of sufficient for the global understanding but does not allow easy reproduction of the results claimed. In my opinion, this is a central point of the paper and it should be fully explained:
-Redundant indexing is used between « V » for vertices and « i » for consecutive angles, whereas they are related. Furthermore, the precise relation with transfer functions previously derived should be explicit (which transfer, which creases etc.)
- $z = \sum(z_v)$ is incorrect since it defines a scalar and not a vector of equations.
-the notation c_i should be made consistent with the authors choice C_α (double stroke font) in the « vertex transfer function » section.
-The paragraph just before « Zeroth loop equation » describing the induction process: « a matrix » : it should be stated that this matrix corresponds to the highest order derivatives. Similarly « this matrix » let the reader think that there is only one matrix (the same) to make singular at each order...
- dz_a/dc_i : Here z_a should be written z_v to comply with the vertices indexing introduced previously.
-page 4: last §: c_α means c_i or something else?
-Just before equation 12: « the \tilde{c} that achieves \tilde{z} » is fuzzy
_Sup Note p8, first sentence: this is Kawasaki-Justin's theorem (Justin actually found it independently before Kawasaki)

Reviewer #4 (Remarks to the Author):

The authors provided the detail responses, and the flow of the manuscript is improved so that the readers can follow it easily. Also, their revised manuscript contains sufficient explanations for each figures. In addition, the FEM analysis to examine face bending behaviors (as shown in Fig. 5 and Figure S.1) is very helpful to support the authors' work, i.e., their simple diagonal crease model. Therefore, I would recommend publication of the revised manuscript in Nature Communications.

Reviewer 2 (Remarks to the Author):

The authors have addressed all my comments and concerns from the earlier draft. I now recommend the paper for publication in Nature Communications.

We thank the reviewer for pointing out relevant citations we had missed and thus helping improve our paper.

Reviewer 3 (Remarks to the Author):

The main text of the paper was indeed improved and the section regarding entropy presents new and potentially interesting results. However, some important issues need to be addressed. Furthermore, technical information is still too fragmented in the supplementary material and there are still many imprecisions and vague justifications.

Hence I suggest a major revision of the paper.

We thank the reviewer for re-evaluating our manuscript and noting our new and expanded results on the entropy of folding modes. We believe our revised manuscript is better motivated and positioned within existing research. In formatting the technical parts of the manuscript, we aimed to make the paper accessible to researchers in physics and materials science in addition to the more mathematically inclined readers.

In keeping with the reviewer's suggestion, we have made further changes documented below.

General comments and questions:

In the section Entropy energy relationship, the authors put the stress on the actuation energy of the self-folding system they investigate. However, the influence of the folds own bending energy (which is must be present compared to the vast range of energies under consideration) is never discussed. One expects that this "folds bending" energy sets a lower bound to the total self folding energy which may not be resolved with higher order loop equations. Does this change the analysis (regarding the "Goldilocks" energy range for instance)?

Indeed, crease folding energy would set a simple lower bound on how soft our folding modes can be made. We have now noted such a lower bound on energy:

"Note that if the creases themselves have non-zero folding energy (e.g., due to finite thickness), the folding energy E_{designed} would be bounded from below by such an energy scale; crease patterns cannot be made softer than the intrinsic stiffness of individual creases."

We note that our COMSOL simulations account for such finite crease folding energy.

In relation to the previous question, are there indeed real physical systems which actually comply with the assumptions made in the paper (rigid folding, no bending energy along the fold, disordered patterns, actuation energy)? (Note that practical systems which are designed to be self-foldable would not be disordered.) If yes, this does not seem clear in the paper (many references are given but are often a bit far from the exact framework under consideration). Describing precisely such systems would give more importance to the entropy relation the authors derived and may suggest its confrontation to experimental evidence.

We have listed numerous origami applications in our paper and explained how, as the reviewer notes, any real material necessarily violates idealizations present in any model. Hence we ran finite element simulations in COMSOL (e.g, see Fig 5 and Fig S.1.) to show that our results are valid even in the presence of complications that arise in real materials such as finite thickness, lack of rigid folding, stretching etc. We have also supplemented these simulations with theoretical arguments based on classic works in the physics literature on crumpling (see Page 3 of manuscript).

Note that our COMSOL simulations used parameters reflecting existing applications of origami based on materials like NiTi sheets and hydrogels, all of which are cited in our manuscript.

For e.g., see discussion of thickness of materials on Page 3:

"We later check the validity of this thin sheet approximation using finite element simulations in COMSOL (see Fig.5 and Supplementary Figure S.1.). Thickness in real application varies, e.g., 0.05 mm thick NiTi sheets of width 50mm for stents [Kuribayashi] to 1 um thick GaAs sheets of width 100 um [ZanardiOcampo] for optically actuated mirrors."

See also discussion after Fig 5 and the Supplementary Figure S.1.

Finally, we do think disordered geometries are critical for future applications of origami since they can achieve geometries and mechanical properties unattainable with periodic Miura-Ori. As we emphasize throughout the paper, existing origami applications already range widely in geometries needed, energy and length scales; future applications are only likely to expand these regimes. Consequently, we think our work that systematically explores all energy regimes is well motivated. For the same reason, it is critical to study disordered patterns and understand the full space of realizable geometries and corresponding mechanical constraints for practical applications. We do not think one can make assumptions about that future applications will only require specific periodic Miura-like geometries, or specific energy or length scales.

Fig1-c and d are still too close to some classical rigid foldable creases patterns. Roughly speaking, showing only that node shaking still gives almost rigid-foldable patterns is not very surprising and mitigates the interest of solving higher-order loop equations. It would give much more strength to the paper if the authors exhibit a kind of random pattern which presents the attracting property of being quasi rigid foldable. The corresponding video is the one to be in the supplementary materials. Possibly this pattern may present several folding path with different energy scales which would also be present in the video.

While Fig 1.c,d may superficially look similar to rigid foldable crease patterns, they are in fact are very different from such patterns. For example, 'node shaking' suggested by the reviewer will violate all loop equations and give much stiffer patterns than those shown. In contrast, our patterns in Fig 1.c,d, and now e exactly solve a number of loop equations and thus are significantly softer than patterns with randomly displaced vertices. Hence we believe all three patterns - Fig 1c,d and now 1e - do fit the reviewers description of 'quasi-foldable'. As we note in the figure's caption explicitly: *"These patterns solve exactly only one (d) or two (c) loop equations."*

However, we understand this point might be missed by a casual reader who glances at Fig 1.

Hence as suggested by the reviewer, we have now added the photograph of a completely disordered origami pattern solving one loop equation to Fig 1. See panel Fig 1e.

The discussion about the number of independent loop equations compared to the number of design parameters is very interesting and should be justified/investigated further (currently there are only assertions and empirical testing in the sup. note)

We thank the reviewer for showing interest in the mathematical nature of the loop equation expansion. In the SI, we have explained why one expects only 5 loop equations based on an alternative approach of Tachi's and that we numerically verified that 5 of our loop equations are indeed independent. However, an analytic exploration of the high order loop equations would not add anything to the main thrust of the paper - an energy-scale dependent exploration of origami patterns.

Hence we think such detailed algebraic work about rigid foldable patterns is beyond the scope of this paper.

The use of *we* is excessive in the whole manuscript (especially in the supplementary notes) and should be made only when it is a conceptual choice of the authors. Otherwise the passive form is perfectly suitable (typically when talking about simple facts or direct consequences). This excessive use of *we* gives the feeling that the authors are giving themselves credits for already known results:

our loop equations, low order loop equations are already known.

we find that the first loop equation: this is already known

our strategy: the expansion of the non-linear loop equation (yet tedious) is actually classical in perturbation theory. etc.

Being mixed with already known results this does not shed enough the light on what is actually new in the paper (going higher order, entropy etc.)

The reviewer is thanked for the helpful remarks on use of the active voice. We certainly did not mean to imply a novel contribution by each use of the first person 'We' (such as when 'we expand this function as a series'). Researchers in many fields prefer writing in active voice as a less cumbersome style of writing.

We understand that in some cases the passive form is more appropriate for our manuscript and have made such changes:

In the introduction,

"How do we specifically design..." is changed to *"How to design..."*.

Furthermore, *"We present three primary results"* is changed to *"This work introduces..."*.

In the discussion,

"our loop equations" was changed to *"the loop equation hierarchy"*.

In SI Note 1,

"Our strategy..." was changed to *"In the spirit of perturbation theory..."*

In SI Note 2,

"we find that the first loop equation" was changed to *"the first loop equation"*.

Detailed Comments:

-Videos are missing in the material transmitted to the reviewer.

We did upload the videos again with the last submission; we will make certain that all supplementary information, including videos, will be provided with the revised manuscript.

-Main Text. p3 first sentence; edges are too soft : why using the fuzzy word soft whereas the text in the parenthesis is simple and accurate?

We thank the reviewer for the close reading of our manuscript. Throughout our paper, we strove to strike a balance in our presentation between rigor for mathematicians and readability for a broader audience.

We think the contrast between ‘rigid’ and ‘soft’ in that paragraph makes the text more intuitive and readable for a wider audience. As a compromise, for the more mathematically minded, ‘soft’ is immediately defined by the parenthetical statement.

-Main Text, equation 3: it is conventional that three lines means definition . = is preferred.

We agree with the reviewer. We have now replaced ‘definition’ sign with an equality.

-Main Text, three after equation 5: augmented quads should be properly defined.

We thank the reviewer for noting this oversight. We have now defined it in the text as

”a quad with an additional face diagonal crease”.

-Main Text, p5 3: The word naturalness sounds awkward. Refer simply to the class of a MV sequence.

-Sup. Note, page 3, 1: replace enveloping by adjacent

We have now made these terminology changes as the reviewer suggested.

-Main Text, p5 4: the normalization with ρ_{crease}^2 is missing in the definition of $\log E$.

We have now changed the definition to reflect the existence of this constant; its value does not affect the derived entropy scaling law since the normalization of ρ^2 becomes an additive constant in log space.

-Fig 3b: name of the horizontal axis: the square seems to apply to the whole ln function which is ambiguous.

We have now fixed this ambiguity by introducing parentheses.

-Main Text p6 3: what is plaquette ?

We thank the reviewer for noting that the term plaquette might not be easily understood from the text. It is changed to *”inner face”*.

-Main Text p6 4 after equation 7: the authors refer to the Sup Note 2, but the evoked details do not seem present in the note.

We thank the author for pointing out this oversight. We did not mean to include this remark as we believe a lengthy discussion about this topic leads the reader away from the main flow of our manuscript. The reference is removed in the revised manuscript.

-Sup. Note, Figure page 3: the symbols are too small

-Sup. Note, First Figure page 4: the symbols are too small

We have now enlarged the symbols in the figures of the revised manuscript.

-Sup. Note, page 4, 1, last sentence. The correction mentioned in the authors answer to me is not present. Furthermore, the sentence itself is still difficult to understand: R is corresponding to which creases and which case on the figure (which is barely readable)? Finally, this is where the choice of MV comes to place: this should be linked with the MV class in the main paper and explained to which extent all possible choices were investigated in the paper.

We thank the reviewer for pointing out this oversight. The clarification now appears in the revised SI:

"One intuitive explanation is that if the angles at the vertex were all 90 degrees, R would be infinite if ρ_α, ρ_γ have opposite signs. The choice of branch for each vertex around the quad sets the MV class of the quad, as discussed in the main text."

-Sup. Note, page 4, Strategy : Notations are still confusing in spite of previous request. They may be clear for the author and kind of sufficient for the global understanding but does not allow easy reproduction of the results claimed. In my opinion, this is a central point of the paper and it should be fully explained: $-z = \text{sum}(z_v)$ is incorrect since it defines a scalar and not a vector of equations.

-Redundant indexing is used between V for vertices and i for consecutive angles, whereas they are related. Furthermore, the precise relation with transfer functions previously derived should be explicit (which transfer, which creases etc.)

-the notation c_i should be made consistent with the authors choice C_α (double stroke font) in the vertex transfer function section.

-The paragraph just before Zeroth loop equation describing the induction process: a matrix : it should be stated that this matrix corresponds to the highest order derivatives. Similarly this matrix let the reader think that there is only one matrix (the same) to make singular at each order

$-dz_a/dc_i$: Here z_a should be written z_v to comply with the vertices indexing introduced previously.

-page 4: last : c_α means c_i or something else?

-Just before equation 12: the \dot{c} that achieves \ddot{z} is fuzzy

We thank the reviewer for his/her many useful comments about the derivation of the loop equations. We have now revised this mathematically complex part of our manuscript along the lines suggested by the reviewer.

1. We have now assigned the index a for vertices around the loop. The cosine notation is made consistent by setting $C_i \equiv \cos(\rho_i)$.

2. As the reviewer points out, $\mathbf{z}_a = 0$ is a vector equation. $\mathbf{z} = \sum_a \mathbf{z}_a$ is thus a sum of vector equations (of the same dimension), and hence a vector equation as required. Using this notation $d\mathbf{z}_a/dc_i$ is now consistent with all other indices.

3. We have now clarified our remarks about the induction process using matrix equations, emphasizing matrices correspond to the highest order derivatives, e.g. *"we must make a matrix corresponding to the highest derivatives singular"*.

4. The confusing remark before equation 12 was changed to the more accurate *"If the first loop equation is satisfied, the solution $\dot{\mathbf{C}}$ that satisfies equation (9) obeys..."*

-Sup Note p8, first sentence: this is Kawasaki-Justins theorem (Justin actually found it independently before Kawasaki).

We thank the reviewer for pointing out Justin's contribution. The text was changed to reflect the contributions of both Kawasaki and Justin.

Reviewer 4 (Remarks to the Author):

The authors provided the detail responses, and the flow of the manuscript is improved so that the readers can follow it easily. Also, their revised manuscript contains sufficient explanations for each figures. In addition, the FEM analysis to examine face bending behaviors (as shown in Fig. 5 and Figure S.1) is very helpful to support the authors work, i.e., their simple diagonal crease model. Therefore, I would recommend publication of the revised manuscript in Nature Communications.

We thank the reviewer for his/her comments and suggestions to expand the scope of our paper through finite element simulations and direct connections to different origami applications. We are happy the reviewer now recommends our paper for publication.